# Calibration of global MODIS cloud amount using CALIOP cloud profiles

Andrzej Z. Kotarba[1]

[1]Space Research Centre, Polish Academy of Sciences, 00-716 Warsaw, Poland

*Correspondence to*: Andrzej Z. Kotarba (akotarba@cbk.waw.pl)

**Abstract.** The Moderate Resolution Imaging Spectroradiometer (MODIS) cloud detection procedure classifies instantaneous fields of view (IFOV) as either 'confident clear', 'probably clear', 'probably cloudy', or 'confident cloudy'. The cloud amount calculation requires quantitative cloud fractions to be assigned to these classes. The operational procedure used by the MODIS Science Team assumes that 'confident clear' and 'probably clear' IFOV are cloud-free (cloud fraction 0%), while the remaining
categories are completely filled with clouds (cloud fraction 100%). This study demonstrates that this 'best guess' approach is unreliable, especially on a regional/ local scale. We use data from the Cloud-Aerosol Lidar with Orthogonal Polarization (CALIOP) instrument flown on the Cloud-Aerosol Lidar and Infrared Pathfinder Satellite Observation (CALIPSO) mission, collocated with MODIS/ Aqua IFOV. Based on 33,793,648 paired observations acquired in January and July 2015, we conclude that actual cloud fractions to be associated with MODIS cloud mask categories are 21.5%, 27.7%, 66.6%, and 94.7%.
Spatial variability is significant, even within a single MODIS algorithm path, and the operational approach introduces uncertainties of up to 30% of cloud amount, notably in polar regions at night, and in selected locations over the northern hemisphere (e.g. China, the north-west coast of Africa, and eastern parts of the United States). Consequently, applications of MODIS data on a regional/ local scale should first assess the extent of the uncertainty. We suggest using CALIPSO-based cloud fractions to improve MODIS cloud amount estimates. This approach can also be used for MODIS/ Terra data, and other
passive cloud imagers, where the footprint is collocated with CALIPSO.

## 1 Introduction

Cloud plays a key role in distributing solar energy in the Earth's atmosphere (Trenberth et al., 2009). Consequently, research into the present and future state of the climate system requires accurate information about cloud amount. Depending on its frequency and physical properties, cloud can both heat (greenhouse effect: +30 Wm$^{-2}$) and cool (albedo effect: –48 Wm$^{-2}$) the
atmosphere. Their net effect on the planetary radiation budget is negative, meaning the Earth would be warmer if all cloud disappeared (Ramanathan and Kiehl, 2006).

The Global Climate Observing System identifies 13 Essential Climate Variables. This set of critical environmental parameters characterize the Earth's climate (Hollmann et al., 2013); they not only include cloud properties, but also highlight that our knowledge of cloud relies largely on satellite remote sensing. Satellite cloud climatology starts with a cloud mask. The aim is

to decide whether cloud is present in a sensor's instantaneous field of view (IFOV), or whether it is cloud free. Input data includes at-sensor registered radiances, along with other auxiliary information that aims to maximize cloud detection.

Efficient cloud detection algorithms have to consider the technical limitations of sensors, available computing power, and environmental factors such as the background (e.g. water, land, snow) and solar illumination (day and night). The resulting cloud mask takes the form of a map that divides IFOV into at least two categories: 'cloud free', and 'cloud contaminated' (or

'cloud filled'). Many masking algorithms introduce additional categories in order to reflect the level of uncertainty in cloud detection (Derrien and Le Gléau, 2005; Dybbroe et al., 2005; Kopp et al., 2014).

The Moderate Resolution Imaging Spectroradiometer (MODIS) is a cloud imaging instrument that is flown onboard NASA's polar orbiting satellites: Terra and Aqua. Circling the Earth in the morning orbit (10:30 local solar time; Terra), and afternoon orbit (13:30 local solar time, Aqua), these twin sensors provide a global picture of cloud four times each day, at 1 km/pixel

resolution (Guenther et al., 2002; Platnick et al., 2003). With 36 spectral channels, continuous correction for orbital drift, and precisely-calibrated detectors, MODIS has set a new standard in cloud remote sensing, and is still considered to be a state-of-the-art cloud imager, despite being launched in 1999 (Terra), and 2002 (Aqua).

MODIS's cloud detection scheme results in four cloud mask categories: 'confident cloudy', 'probably cloudy', 'probably clear', and 'confident clear' (Frey et al., 2008). The fact that these classes are presented as qualitative, text-based labels rather

than a numeric probability causes the technical problem of how to quantitatively interpret these labels. A numeric interpretation is mandatory when instantaneous observations (Level 2 products) are aggregated spatially and/ or temporally to provide climatological information such as mean monthly cloud amount (Level 3 products).

The procedure implemented by NASA's MODIS Science Team (hereinafter the 'standard' or 'operational' procedure) is to assume that IFOV declared 'confident cloudy' and 'probably cloudy' are, in fact, 100% cloud filled, while 'confident clear'

and 'probably clear' are completely cloud free (cloud fraction of 0%) (Hubanks et al., 2008). The approach is widely used whenever there is a need to make a binary distinction between cloudy and cloud-free pixels – e.g., Gao et al. (2008), Remer et al. (2012), Wilson and Oreopoulos (2013), Wilson, Parmentier, and Jetz (2014), Kraatz, Khanbilvardi, and Romanov (2017), Gomis-Cebolla, Jimenez, and Sobrino (2020).

However, since the MODIS Science Team (ST) approach is only a 'best guess', alterative assumptions are also used. For

instance, it can be assumed that only 'confident cloudy' pixels are 'cloudy', while all remaining classes are 100% cloud free. Similarly, only 'confident clear' detections can be considered as truly cloud-free, while all other classes are assumed to be 100% cloud filled (Li et al., 2005). Krijger et al. (2007) argue that the latter approach leads to the false detection of small clouds, while cloud is frequently overlooked if the first method is applied. Another approach is simply to exclude 'probably clear' and 'probably cloudy' detections from the analysis. This strategy was adopted by Chan and Comiso (2013), whose work

was based on only 'confident clear' and 'confident cloudy' categories of MODIS data.

Quantitative studies have shown that only considering the 'confident cloudy' class as cloudy may be more consistent with other cloud data such as Landsat observations (Melchiorre et al., 2020), or visual observations at meteorological stations

(Kotarba, 2015). On the other hand, Fontana et al. (2013) compared MODIS data with ground-based observations in Switzerland (4 stations, 12 years of data), and found that results varied from station to station.

The theoretical range of uncertainty related to various interpretations of the MODIS cloud mask was investigated by Kotarba (2015). The latter study found that the global cloud amount estimates may differ by up to 14%, depending on whether only 'confident cloudy' detections are considered to be 'cloudy', or whether the definition is extended to include intermediate classes. The discrepancy was found to increase by up to 40–60% regionally, suggesting that MODIS cloud estimates are very uncertain in these areas. Such a wide range of uncertainty makes it difficult to run reliable studies on the climate system.

Neither the MODIS ST standard procedure, nor any other 'best guess' variants have been validated on a global scale. Most importantly: no research-based, objective alternatives to the 0/0/100/100 interpretation have been suggested. This study addresses this problem. Specifically, it provides global cloud fractions based on quantitative analysis of CALIOP lidar observations.

CALIOP (the Cloud-Aerosol Lidar with Orthogonal Polarization) is a cloud profiling instrument flown onboard the CALIPSO
(Cloud-Aerosol Lidar and Infrared Pathfinder Satellite Observation) spacecraft. Launched in 2016, CALIPSO flies in close formation with the Aqua satellite, therefore both instruments – MODIS and CALIOP – sample the same fragment of the atmosphere tenths of seconds apart (Stephens et al., 2018). In this study, CALIPSO data is considered as ground truth. This is because CALIOP is an active remote sensing instrument, which means that it can sample the atmosphere during the day and at night with comparable sensitivity. Imaging radiometers (such as MODIS) perform less effectively at night, when solar
channels are missing. Furthermore, the use of short wavelengths makes CALIOP very sensitive to cloud of low optical thickness (e.g. sub-visual cirrus) that is often missed by imagers (Ackerman et al., 2008).

In the following sections we seek to answer the questions: 1) What quantitative cloud fractions (based on CALIOP observations) should be applied to MODIS thematic cloud mask classes? and 2) What uncertainties in global cloud amount are introduced by the MODIS ST standard procedure? Finally, we evaluate whether the standard procedure for calculating the
MODIS Level 3 cloud amount is reliable.

## 2. Data & Methods

### 2.1 MODIS data

The MODIS cloud detection scheme is based on thresholds that are applied to brightness temperature (thermal bands), and reflectance (solar channels), derived from observations in 22 spectral bands ranging from 0.66 µm to 13.9 µm. Ackerman et
al. (1998), Frey et al. (2008), and Baum et al. (2012) provide very detailed descriptions of the cloud masking procedure. The general concept is as follows.

The algorithm executes a series of tests, each of which results in a confidence level (ranging from 0 to 1) that a particular IFOV is cloud free. Tests to detect similar cloud types are grouped. The lowest confidence level for a test within a group is set as the confidence level for the whole group. Confidence levels for groups are then multiplied to determine the final confidence

level (Q). Following this procedure, the IFOV is assigned to one of four cloud mask classes: 'confident clear' ($Q > 0.99$), 'probably clear' ($Q > 0.95$), 'probably cloudy' ($Q > 0.66$), or 'confident cloudy' ($Q \leq 0.66$). Numeric values of the confidence level for an individual test, a particular group of tests, and the final confidence level (Q) are not provided within the MODIS Cloud Mask product, despite being potentially useful for a quantitative interpretation of Cloud Mask thematic categories. Only these four thematic classes are reported, and used for further processing of MODIS cloud data.

The exact number of spectral tests executed for an IFOV varies from a few to over a dozen, depending on the path through the algorithm. Paths reflect different environmental conditions, and are introduced to maximize success. Dedicated sets of spectral tests are executed for land, ocean, desert and coastal areas, for both day and night conditions. The presence of snow and/ or ice is taken into account, as is sunglint over oceans. Separate thresholds have been introduced for polar regions, which are defined as land and ocean within 30 degrees of each pole.

Cloud detection results are stored in the 48-bit 'Cloud Mask' product, codenamed MYD35 (Aqua) and MOD35 (Terra) following the MODIS nomenclature. In this study, we evaluated the latest version of MYD35 (Collection 061) data, available in the form of 5-minute granules, at 1 km per pixel spatial resolution (at nadir), with native satellite projection. Each MYD35 file is accompanied by a MYD03 'Geolocation file' product, that stores longitude and latitude information for individual cloud mask IFOV.

**2.2 CALIOP data**

    CALIOP operates at 532 nm and 1064 nm. The instrument's pencil-like beam only scans locations along the satellite's ground track, as a trade-off for information on the vertical structure of cloud/ aerosols. Its spatial resolution is a function of the satellite's altitude. Resolution is finest – 0.333 km horizontal, 30 m vertical – in the troposphere, up to 8.2 km. Between 8.2 km and 20.2 km, vertical resolution falls to 60 m, and horizontal sampling to 1 km. Between 20.2 km and 30.1 km, data are even

coarser: 1.667 km horizontal and 180 m vertical resolution. Higher in the atmosphere (30.1 km to 40.0 km) horizontal resolution decreases to 5 km, while vertical resolution is 300 m (Hunt et al., 2009; Winker et al., 2006).

    CALIOP detects cloud by applying thresholds to 532 nm attenuated scattering ratios. The aim is to separate the cloud signal from the clear air background (molecular scattering), aerosols, and instrument noise. The algorithm calculates cloud base height, cloud top height, and – as a consequence – the number of cloud layers within a profile. Up to 10 layers can be reported.

The procedure is fully automatic (Vaughan et al., 2009). The output is stored in the Level 2 Cloud Layer Data product, available at 333 m, 1 km, and 5 km along-track sampling intervals. Here, we use the 1 km interval (version 4.20; CAL_LID_L2_01kmCLay-Standard-V4-20), as its resolution matches the spatial resolution of the MODIS cloud mask. Furthermore, 1 km is the highest available level of detail for CALIOP data within the troposphere.

    In order to use the CALIOP product to evaluate MODIS data, 3-dimensional cloud layer data was reduced to column-

integrated, binary cloud/ no cloud information. Specifically, we focused on the 'Number_Layers_Found' parameter provided in the CAL_LID_L2 product. 'No cloud' was recorded when the latter variable was set to 0 (i.e. zero layers found), and 'cloud'

otherwise (i.e. at least one layer was reported). Geolocation was based on longitude and latitude arrays included in the product at 1 km spatial resolution.

In some cases, cloud and aerosol can appear similar to CALIOP. The cloud-aerosol discrimination (CAD) score, which is a numerical index stored in the CAL_LID_L2 product, provides information about the algorithm's uncertainty in separating cloud and aerosol. In the case of cloud, CAD values vary between 0% (it is unclear whether aerosol or cloud was observed) and 100% (cloud detected with the highest confidence). The index is calculated for each cloud layer found in the CALIOP atmospheric profile. Since our study focuses on column-integrated information of cloud presence, we selected the highest CAD value within a profile. Statistics for January and July 2015 showed that 95.6% of considered CALIOP observations were characterized by a CAD score of at least 70%, while it was below 20% for only 1.5% of data. Therefore, the selected CALIOP data can be considered as a reliable reference for MODIS. See Supplementary Online Materials, Fig. S1 for more detailed statistics about CAD scores.

## 2.3 Matching CALIOP and MODIS data

Matching CALIOP data is a well-established method for the calibration/ validation of atmospheric products from various space missions. It has already been widely used for MODIS/ Aqua (Baum et al., 2012; Holz et al., 2009; Sun-Mack et al., 2014; Wang et al., 2016; Xie et al., 2010), and sensors flown onboard Suomi-NPP, NOAA, and MetOp polar orbiting spacecraft, which occasionally synchronize their orbital configuration with CALIPSO (Hutchison et al. 2014; Heidinger et al. 2012; Karlsson and Johansson 2013; Karlsson and Dybbroe 2010). CALIPSO also passes within the field of view of other geostationary satellites, and CALIOP data is used to assess their atmospheric products (Sèze et al., 2015; Shang et al., 2018). In this study, Aqua/ MODIS data for January and July 2005 were paired with corresponding CALIPSO/ CALIOP observations. The matching procedure selected a MODIS IFOV and compared it with the corresponding CALIOP profile (where the geometric centre was within the selected MODIS IFOV). The orbital configuration of the two missions only allows CALIOP to sample MODIS IFOVs that are close to the MODIS nadir (due to low sensor viewing angles). Consequently, matching observations across the whole width of the MODIS swath is not possible.

Although very straightforward, the procedure was time-consuming since a single MODIS granule contains ~2030 IFOV, and a full day of Aqua observations produces 288 granules. The final database consisted of 33,793,648 MODIS–CALIOP paired observations. Average spatial separation between the centres of MODIS and CALIOP IFOV was 418 m, and 19% had a separation of less than 250 m. Temporal differences between lidar and imager observations were determined using the spacecrafts' on-orbit separation, and ranged from 60 sec to 97 sec (81 sec on average). Our dataset excluded one MODIS cloud mask processing path: sunglint. This was because CALIPSO's orbit has been intentionally designed to avoid sunglint areas, in order to avoid the lidar being 'blinded' by solar reflection from the ocean.

Our empirical calculation of cloud fraction in each MODIS cloud mask class was based on the ratio of CALIOP cloudy detections to all detections within a class. A perfect MODIS cloud detection algorithm would categorise a 0% cloud fraction as 'confident clear', while a 100% cloud fraction would be categorised as 'confident cloudy'.

The resulting MODIS-CALIOP statistics have been spatially aggregated, and are shown as global maps with 2.5°×2.5° longitude and latitude resolution.

## 3 Results

### 3.1 Misdetection of cloud and clear sky by MODIS

The first key point that emerged from the matched MODIS–lidar observations was the accuracy of MODIS cloud detections.
Overall accuracy in January and July 2015, compared to reference CALIOP data, was 89.4% during the day, and 84.2% at night (Tab. 1). This statistic assumes that 'probably clear' detections are merged with 'confident clear', and 'probably cloudy' detections are combined with 'confident cloudy'. If a less tolerant approach is applied, i.e. only 'confident clear' and 'confident cloudy' detections are considered ('probably clear'/ 'probably cloudy' classes are interpreted as misdetections), overall accuracy fell to 81.9% during the day, and 73.3% at night.

Clouds missed by MODIS, but detected by CALIOP were most frequent during the polar night, regardless of the hemisphere (Fig. 1c, d). Up to 40% of MODIS 'confident clear' and 'probably clear' detections were found to be incorrect around Antarctica in July, and the Arctic in January. Globally, daytime (Fig. 1a, b) misdetections were around half of those at night. They only exceeded 30% locally, and polar regions were less affected. A notable observation was July in the northern hemisphere, where only a few small regions of misdetection were observed. The analysis highlighted a belt of relatively higher
frequency misdetections (15–25%) in the equatorial zone; here the magnitude of the effect was similar for both day and night. Only a few occasions were identified when over 10–15% of MODIS 'confident cloudy' and 'probably cloudy' observations were identified as cloud-free by CALIOP (Fig. 2). Further analysis showed that although false detection was rare in polar regions, it was significant in specific regions of the northern hemisphere. North-east China emerged as the most problematic area (Fig. 2a). Here, 50–70% of MODIS 'confident cloudy' and 'probably cloudy' detections were cloud-free according to
CALIOP. However, this high rate of false detection was only observed in January, and only during the day.

### 3.2 Cloud fraction for cloud mask classes

Our empirical calculation of cloud fraction in each MODIS cloud mask class was based on the ratio of CALIOP cloudy detections to all detections within a class. A perfect MODIS cloud detection algorithm would categorise a 0% cloud fraction as 'confident clear', while a 100% cloud fraction would be categorised as 'confident cloudy'.

On average, one fifth of MODIS 'confident clear' detections were found to be cloudy by CALIOP. Consequently, the average cloud fraction for this class was 21.5%, instead of the theoretically expected 0% (Tab. 2). At night, the fraction was over twice the daytime value (29.5% compared to 12.7%). On the other hand, pixels flagged by the MODIS algorithm as 'confident

cloudy' were, almost always, contaminated with some cloud, and were sometimes cloud-filled. Regardless of the time of day, the actual CALIOP-based cloud fraction for 'confident cloudy' detections was close to 100%, reaching 94.7%.

MODIS intermediate classes constituted 13.3% of all detections. CALIOP cloud fractions were 27.7% and 66.6% for 'probably clear', and 'probably cloudy' classes respectively. The statistics revealed a difference of up to 17% between day and night conditions, and it was especially small for the 'probably clear' class (1.3%) and the 'confident cloudy' class (daytime had no impact at all).

   The parameters reported in Table 2 are global averages (means), and spatial diversity was observed. Differences were smallest
for the 'confident cloudy' class – both during the day (Fig. 3a) and at night (Fig. 3b) – and the CALIOP-based cloud fraction exceeded 90% at almost every location. North-east China, the southern Arabian Peninsula and Eastern Antarctica were the only significant exceptions; here cloud fraction decreased to 50–70%.

   The cloud fraction distribution was homogeneous for the 'confident clear' class, however, only during the day (Fig. 3g). At night (Fig. 3h) it increased substantially in polar regions, especially over the oceans of the southern hemisphere, along the
coast of Antarctica. Unlike polar regions, no noticeable day/ night difference was observed for mid- and low-latitudes (< 10%); here, the cloud fraction was very low (< 20%), and very few MODIS 'confident clear' detections were identified as cloudy by CALIOP.

   Among the MODIS intermediate classes, 'probably clear' differed most from CALIOP-based data. First, a high cloud fraction (> 70%) was observed at night along the equator and in polar regions (both oceanic and continental; Fig. 3f). At mid-latitudes
the cloud fraction for MODIS 'probably clear' observations was relatively low (< 10–20%). This pattern was inverted during the day (Fig. 3e). At this time a higher (50–75%) cloud fraction was noted for mid-latitudes, and over parts (typically land) of the polar regions.

### 3.3 Cloud fraction as a function of the algorithm path

   The MODIS cloud detection algorithm distinguishes between day and night (Tab. 1, Tab. 2), and four types of background
(land, desert, coast, ocean), each of which can be either snow-covered or snow-free. CALIOP-based cloud fractions for all algorithm paths are reported in Table 3. These values give a detailed understanding of MODIS cloud detection results. Data are given for each class of the MODIS cloud mask separately. In our study, we structured the paths through the algorithm in more detail. Snow-covered conditions were considered for land, desert, ocean and coast separately, while in the MODIS algorithm they are grouped as snow/ ice. This greater level of detail allowed us to observe how the presence of snow impacted
the cloud mask over different backgrounds.

   Per-class estimates of cloud fraction were very consistent for all algorithm paths for the 'confident cloudy' category (Tab. 3). Final values ranged between 97.7% (night, snow-free, land) and 86.4% (night, snow-covered, desert), and were close to the standard Level 3 assumption of 100%. This finding contrasted with cloud fractions found for the 'confident clear' category. While MODIS recorded cloud-free conditions, CALIOP data revealed that the actual cloud fraction ranged from 8.0% (night,
snow-free, land) to 49.7% (night, snow-covered, ocean).

The combination of night, an oceanic background and snow-covered (or sea ice) constituted the 'most cloudy' scenario (Fig. 4). Here, a very high cloud fraction was found for not only the 'confident cloudy' category (96.8%, Fig. 4a), but also all remaining classes: 82.5% ('probably cloudy'; Fig. 4b), 73.3% ('probably clear'; Fig. 4c) and, surprisingly, up to 49.7% for 'confident clear' (Fig. 4d).

CALIOP-based cloud fractions were most consistent with the standard Level 3 interpretation for snow-free land at night (Fig. 5). Here, the cloud fraction for 'confident clear' was low (8.0%; Fig. 5d), and very high for 'confident cloudy' (97.7%; Fig. 5a). At the same time, intermediate classes were well-separated: 68.5% for 'probably cloudy' (Fig. 5b), and 25.6% for 'probably clear' (Fig. 5c). Globally, no significant difference was found for cloud fraction values for the night/ snow-free/ land algorithm path. A small exception was noted for the 'probably clear' type, where the cloud fraction was 10–30% higher in the 230 tropics compared to the rest of the world.

A similar spatial distribution of CALIOP-based cloud fractions was observed for snow-free land during the day – the scenario of particular interest for land/ vegetation remote sensing with MODIS. The two notable differences were related to 'probably clear' (Fig. 6c) and 'confident clear' categories (Fig. 6d). The latter occurred twice as often during the day (15.6%) than at night (8.0%). Similarly, cloud was more frequent in the 'probably clear' class. However, this was only found in the tropics and 235 at high latitudes, which mirrored a zonal pattern that was only weakly seen at night.

As ice-free oceans represent the majority of Earth's surface, cloud detection over ocean is the most frequent algorithm path. Daytime conditions make detection easier (due to the availability of solar channels). Under such circumstances, CALIOP detected cloud in 10.5% of MODIS's 'confident clear' observations (Fig. 7d), and confirmed 95.2% of 'confident cloudy' detections (Fig. 7a). Cloud fractions for intermediate classes (daytime over ice-free ocean) were 54.5% and 28.4% for 240 'probably cloudy' (Fig. 7b) and 'probably clear' (Fig. 7c) categories, respectively. 'Probably clear' was the only class where there was a clear latitude-dependent cloud fraction: values increased by 30–60% along a path ~30–40 degrees north/ south.

## 4 Discussion

Our investigation of spatially and temporally collocated MODIS (cloud imager) and CALIOP (cloud profiling lidar) observations for January and July 2015 revealed that MODIS Collection 061 global cloud amount estimates are imperfect. 245 During the generation of the Level 2 product, the masking algorithm fails to accurately report cloud over polar regions, and over selected locations at lower latitudes. Consequently (as discussed in this section) the Level 3 product generation underestimates cloud fractions for cloud mask classes in numerous regions. The reliability of these results depends on several factors, most notably the spatial and temporal accuracy of Aqua/ MODIS and CALIPSO/ CALIOP collocation.

Temporal differences between Aqua and CALIPSO observations varied from 60 seconds to 97 seconds. In this time, cloud can 250 develop and move, introducing the risk that CALIOP observes a different state of the atmosphere compared to MODIS. Várnai and Marshak (2009) evaluated the problem by comparing MODIS reflectance with that collected by the Wide Field Camera. The latter is an imaging instrument flown onboard CALIPSO, along with CALIOP. They found that for low cloud, radiance

differed only slightly over 72 seconds, and it was reasonable to ignore any discrepancies when focusing on aerosol properties (they gave no particular conclusions for cloud). In order to test how sensitive our results were to the time shift, we calculated

the overall accuracy of the cloud detection algorithm as a function of the time between Aqua and CALIPSO passes. The results were very consistent: despite the shift, accuracy remained at 86.7±0.1%. This finding confirmed that the temporal separation between Aqua and CALIPSO had no significant impact on the results of our study.

Another potential source of uncertainty is the geometric mismatch between MODIS and CALIOP IFOV. They are not aligned perfectly: 66% of collocated IFOV were separated by less than 0.5 km, and 82% by less than 0.6 km. Similar statistics – 75%

and 93%, respectively – were found by Wang et al. (2016) in their investigation of cloud based on MODIS and CALIOP observations. To investigate whether geometric conditions did have an impact on our results, we calculated the overall accuracy of the MODIS cloud mask as a function of the distance between MODIS and CALIOP IFOV. For ranges up to 1 km with a 100 m step, the change in accuracy was insignificant: 87.0±0.3% on average. (See Supplementary Online Materials' Fig. S2 for more detailed statistics about the spatial and temporal separation between MODIS and CALIOP).

It is possible that agreement between MODIS and CALIOP data is affected by cloud optical thickness ($\tau$) or, more precisely, by the higher sensitivity of CALIOP in detecting optically-thin cloud. Ackerman et al. (2008) estimated the MODIS limit for $\tau$ to be approximately 0.4. A similar improvement in agreement with CALIOP as a consequence of increasing $\tau$ was observed by Karlsson and Håkansson (2017) for the Advanced Very High Resolution Radiometer (AVHRR) instrument. The latter study demonstrated that the imager's probability of detection changed in the range $0.0<\tau<1.0$. We calculated the same statistic, and

found that the probability distribution for MODIS was identical to AVHRR – although MODIS values were higher. This finding strongly suggests that cloud thickness has the same impact on our results as that found in previous studies. Collection 006 of MODIS data was investigated by Wang et al. (2016), who used lidar–radar (CALIPSO–CloudSat) profiles to focus on daytime multi-layered clouds. Our findings are consistent with those reported by Wang et al. (2016), despite the fact that the latter authors used a dataset of 267 million IFOV, while our study relied on around 33 million profiles. Their validation of

Collection 006 reliability found overall agreement of 77.8% compared to our study, which found 81.9%. The difference may be due to the different sample sizes. Our result for cloud-free sky detection was slightly higher than in Wang et al. (2016): 25.5% compared to 20.9%. On the other hand, results for cloudy sky detection were very similar: 56.9% compared to 56.4% in our study.

Our study revealed that even for Collection 061, i.e. the most recent (July 2020) version of the MODIS cloud mask, up to 40%

of cloud-free skies detected during the polar night were actually cloudy. Daytime accuracy was lowest over China (in January), the USA/ Canada (in January) and over tropical ocean along the west (January) and east (July) coasts of Africa. In these cases, MODIS detected cloud that did not exist according to CALIOP. False detections may be due to snow cover (the USA/ Canada), high aerosol content over China (Zhang et al. 2019; Tan, Zhang, and Shi 2019), and ocean bordering desert regions in North Africa (Weinzierl et al., 2017; Zuluaga et al., 2012).

As reported by Wang et al. (2016), and previously by Baum et al. (2012) and Ackerman et al. (2008), cloud detection in polar regions remains an unresolved issue for MODIS, and similar passive imaging radiometers. Polar night is especially

challenging. Successful discrimination between cloud and the underlying surface requires radiance measurements in ice absorption bands (e.g. 1.6 µm or 2.1 µm). But as these are only available in daytime, night-time detection has to rely on thermal infrared data. As thermal inversion in the polar tropopause decreases the thermal contrast between cloud and the background,
the thermal signatures of cloud and the land/ ocean surface become indistinguishable, leading to cloud masking errors (Liu et al., 2004). CALIOP, however, does not require solar illumination to operate. As it uses light emitted by the instrument itself, its performance is far less affected by day-night conditions. CALIOP's night-time data are of even higher quality, because solar illumination introduces an additional background signal and, thus, decreases the signal-to-noise ratio (Hunt et al., 2009). Furthermore, MODIS tends to miss up to ~20% of cloud along the Intertropical Convergence Zone (ITCZ), regardless of the
time of year (January or June), and the time of day. This can be partially explained by the fact that MODIS is less sensitive to optically thin cloud than CALIPSO, and the ITCZ is the region where cirrus is most frequently observed (Sassen et al., 2009). The higher sensitivity of CALIOP to optically thin cirrus, and the higher sensitivity of the lidar during night-time, also explains why CALIOP-based cloud fractions for MODIS 'confident clear' and 'probably clear' classes are higher along the ITCZ at night (Fig. 3f, h) than during the day (Fig. 3e, g).

The main goal of our study was to investigate the validity of the standard (operational) approach to the quantitative interpretation of MODIS cloud mask classes. The most important consequence of calculating empirical cloud fractions for MODIS cloud mask categories is the ability to recalculate global cloud amount with new weights. Therefore, instead of using global fractions (reported in Table 2), we derived a set of dedicated fractions for each algorithm path, and each 2.5-degree grid box (i.e. a local equivalent to the data given in Table 3). This considers MODIS IFOV within the full swath (excluding
sunglint), and not only those collocated with CALIOP. Full-swath data were used because the MODIS L3 cloud amount product applies the same cloud mask interpretation to all IFOVs, regardless of their off-nadir angle. On the other hand, the use of nadir-only MODIS observations would result in CALIOP-like spatial coverage of the data, creating significant gaps due to CALIOP's pencil-like viewing geometry. Figure 8 illustrates the results of the calculation and reports differences in cloud amount between the MODIS ST operational product, and the product generated using the fractions presented in this study.

The outcome of the simulation shows that the use of current operational cloud fractions introduces significant errors. In some locations, MODIS underestimates cloud amount by 20–40%, most notably in polar regions at night. An overestimation of similar magnitude is observed mostly over the northern hemisphere: the USA/ Canada and China in January (both day and night), and the tropical coasts of Africa during the day (both in January and July). Consequently, MODIS Level 3 estimates of cloud amount should be used with great caution in those regions. This is especially important for the Arctic, which is
undergoing a rapid change in climatic conditions (Serreze and Barry, 2011), and where cloud has been found to be an essential element in feedback (Kay et al. 2008; Vavrus 2004; Shupe and Intrieri 2004; Tan and Storelvmo 2019).

The availability of collocated MODIS and CALIOP observations also allowed us to examine which of the three 'best guess' interpretations of cloud mask categories is most accurate: the one when only 'confident cloud' IFOV are 'cloudy', the one that only considers 'confident clear' as 'clear', or the operational approach? We therefore calculated merged global cloud amount
for January and July 2015. Our results show that, on the global scale, the standard approach is closest to CALIOP reference

data, although only during the day (Tab. 4). At night, it is more accurate to assume that only 'confident clear' is actually cloud-free. The global result is biased by the polar night. In these conditions, all three 'best guess' interpretations noticeably underestimate cloud amount. At low- and mid-latitudes the standard (operational) approach differs from CALIOP data by ±2%. However, it should be noted that these statistics relate to large areas. As our study shows, regional differences are orders of magnitude larger.

Our study assumed that CALIOP's 'cloudy' IFOV was always completely cloud filled. This assumption is common when interpreting cloud masks based on data from the majority of imaging radiometers flown onboard meteorological and land-imaging satellites. However, studies by Zhao and Di Girolamo (2006), and Kotarba (2010) suggest that this postulate may not be true. Both of the latter studies took advantage of a rare collocation between a meteorological imager (MODIS) and the high-resolution land imager (ASTER) flown onboard the Terra satellite. Nearly 3,000 ASTER IFOV were located within each MODIS pixel. Kotarba (2010) showed that for sunglint-free, oceanic scenes in the tropics, actual cloud coverage for the 'confident cloudy' MODIS category was 79.2% (mean) or 99.8% (median), instead of the assumed 100%. Comparable statistics for CALIOP are not available, as the CALIPSO spacecraft does not carry a high-resolution imager. Given the lack of alternatives, we must accept the hypothesis that 'cloudy' means 100% cloud filled.

## 5 Summary and Conclusion

This study investigated 33,793,648 collocated MODIS (cloud imager) and CALIOP (cloud profiling lidar) observations, acquired in January and July 2015. Our evaluation of the dataset allowed us to answer three, essential questions, related to global estimates of cloud amount resulting from the MODIS/ Aqua mission. These questions are:

1. *What are the actual cloud fractions corresponding to MODIS cloud mask classes?* We found that these fractions are 21.5%, 27.7%, 66.6%, and 94.7%, rather than the MODIS Science Team-assumed values of 0%, 0%, 100% ,and 100% for 'confident clear', 'probably clear', 'probably cloudy', and 'confident cloudy' categories, respectively (Tab. 2). Importantly, we found that the percentage of cloud cover to be assigned to MODIS cloud mask classes varied spatially (Fig. 3), and recommend that global fractions should be avoided, in favour of local alternatives.

2. *How significant are uncertainties in global cloud amount estimates calculated using the MODIS ST operational approach?* We found that uncertainties were up to −30% of cloud amount in the polar regions at night, and up to +30% of cloud amount in selected locations over the northern hemisphere, more frequently during the day (Fig. 8).

3. *Is the MODIS Level 3 standard approach reliable?* Our results showed that when a global cloud amount value is required (day and night, for all latitudes), the standard approach can be considered reliable (Tab. 4). We found that, in this case, it was more accurate than other 'best guess' approaches – namely only 'confident clear' is 'clear' (other classes are 'cloudy'), and 'confident cloudy' is 'cloudy' (other classes are 'clear'). However, on a

regional scale the standard approach fails (Fig. 8). Whenever MODIS cloud amount is estimated regionally or locally it is necessary to assess whether a particular location might be affected by an error of up to ±30%.

Errors and uncertainties related to the generation of the MODIS Level 3 cloud amount product originate in the Level 2 product: the cloud mask (Fig. 1–2 vs. Fig. 8). The cloud detection algorithm is more-or-less accurate depending on environmental conditions, which are approximated as algorithm paths (Tab. 3). However, conditions within paths are not constant (Fig. 4–7): for instance, the same radiance/ reflectance thresholds are applied to Europe, the USA and China, while environmental conditions in these locations are not the same (e.g. different aerosol loads, different aerosol optical properties). The MODIS

Science Team have attempted to discriminate between these conditions. For instance, since Collection 006 the 0.86 μm reflectance test over land considers thresholds that are a function of the Normalized Difference Vegetation Index (NDVI) and scattering angle. Although cloud misclassification is less frequent than in previous Collections, it still occurs, and impacts the degree of uncertainty of L3 cloud amount estimates, as shown in this study.

CALIOP-based estimates of cloud fraction are a robust way to adjust (and correct) MODIS estimates. The method described

in this paper can be used globally, with the exception of sunglint regions (which are not sampled by CALIOP). In these areas 'best guess' findings can, potentially, be applied. CALIOP also does not sample MODIS IFOVs other than those close to nadir, while increasing the sensor's zenith angle impacts cloud amount estimates (Maddux et al., 2010). However, the MODIS ST standard procedure assumes identical cloud fractions for all cloud mask classes, regardless of the viewing angle. The analogical application of CALIPSO-based cloud fractions may still be an improvement over the MODIS ST best guess procedure, as it

is for nadir IFOVs. The potential availability of lidar data for all MODIS zenith angles can further improve the method.

The polar regions benefit most from the new method. Cloud fractions derived for MODIS/ Aqua may be also adopted for MODIS/ Terra, since the two sensors are expected to produce comparable and homogenous records. Moreover, the occasional collocation of the CALIPSO satellite with AVHRR and VIIRS instruments makes it possible to calculate similar cloud fractions for these missions, and produce more reliable cloud climatologies.

**Data availability**

MODIS data are available from the Level 1 and Atmosphere Archive & Distribution System (LAADS) Distributed Active Archive Center (DAAC) at NASA's Goddard Space Flight Center (https://earthdata.nasa.gov/eosdis/daacs/laads). CALIPSO products are available from the Atmospheric Science Data Center (ASDC) at NASA's Langley Research Center (https://eosweb.larc.nasa.gov/).

**Author contribution**

AZK designed the research, carried it out, and prepared the manuscript.

**Competing interests**

The authors declare that they have no conflict of interest.

**Acknowledgements**

The organization and pre-processing of the CALIPSO database was part of a project funded by the National Science Institute of Poland under contract no. UMO-2017/25/B/ST10/01787. Processing of the MODIS-CALIPSO database and the resulting analyses was funded by the Space Research Centre of the Polish Academy of Sciences, as a part of the research theme 'Satellite monitoring of geophysical processes in the atmosphere and Earth's surface, related to climate and light pollution'.

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

**Tables**

**Table 1**. Agreement in cloud detection between MODIS and CALIOP (% of all cases). Overall accuracy (given in brackets) refers to 'confident clear' and 'confident cloudy' detections. In other cases, 'confident clear' and probably clear' were merged, as were 'probably cloudy' and 'confident cloudy'.

| | | **MODIS** | | | | **Overall accuracy** |
|---|---|---|---|---|---|---|
| | | confident clear | probably clear | probably cloudy | confident cloudy | |
| | | *Day+Night* | | | | |
| CALIOP | clear | 22.7 | 5.4 | 1.9 | 3.1 | 86.7% (77.3%) |
| | cloudy | 6.2 | 2.1 | 3.9 | 54.6 | |
| | | *Day only* | | | | |
| CALIOP | clear | 25.5 | 5.1 | 1.7 | 3.2 | 89.4% (81.9%) |
| | cloudy | 3.7 | 2.0 | 2.4 | 56.4 | |
| | | *Night only* | | | | |
| CALIOP | clear | 20.2 | 5.7 | 2.2 | 3.0 | 84.2% (73.3%) |
| | cloudy | 8.5 | 2.1 | 5.2 | 53.1 | |

**Table 2**. Global cloud fractions for MODIS cloud mask classes derived from CALIOP observations ('This study'), and used in the operational MODIS Science Team Level 3 product ('Operational'). Numbers in brackets refer to class frequency ($n =$ 33, 793, 648).

| Source of cloud fractions for cloud mask classes | | Cloud fractions (%) for MODIS cloud mask class (class frequency, % of $n$) | | | |
|---|---|---|---|---|---|
| | | confident clear (28.9%) | probably clear (7.5%) | probably cloudy (5.8%) | confident cloudy (57.8%) |
| Operational | Day+Night | 0.0 | 0.0 | 100.0 | 100.0 |
| This study | Day+Night | 21.5 | 27.7 | 66.6 | 94.7 |
| | Day only | 12.7 | 28. 4 | 58.4 | 94.7 |
| | Night only | 29.5 | 27.1 | 70.7 | 94.7 |


**Table 3**. CALIOP-based cloud fractions for MODIS cloud mask classes, calculated individually for each MODIS algorithm path. Note that more paths are reported here than in the MODIS project. Snow-covered ocean, land, desert and coast constitute a single path in the operational algorithm, while here they are reported individually to highlight how snow impacts the results. The sunglint path is missing as CALIOP does not sample over sunglint areas. Numbers in brackets refer to how frequently (% of $n$) a given algorithm path was executed, $n = 33, 793, 648$.

| Cloud masking algorithm path | | | | CALIOP-based cloud fractions [%] for MODIS cloud mask class | | | |
|---|---|---|---|---|---|---|---|
| | | | | confident clear | probably clear | probably cloudy | confident cloudy |
| Day (47.2) | Snow-covered (5.5) | Land | (0.2) | 13.8 | 67.0 | 56.0 | 97.6 |
| | | Desert | (3.9) | 12.6 | 32.6 | 71.8 | 96.6 |
| | | Coast | (0.2) | 15.3 | 55.5 | 61.8 | 93.8 |
| | | Ocean | (1.1) | 20.5 | 76.3 | 69.7 | 88.6 |
| | Snow-free (41.7) | Land | (6.7) | 15.6 | 32.3 | 63.9 | 93.4 |
| | | Desert | (3.4) | 9.1 | 19.1 | 45.5 | 90.0 |
| | | Coast | (1.6) | 19.0 | 33.8 | 59.8 | 93.0 |
| | | Ocean | (30.1) | 10.5 | 28.4 | 54.5 | 95.2 |
| Night (52.8) | Snow-covered (15.8) | Land | (2.6) | 31.4 | 65.0 | 80.9 | 93.9 |
| | | Desert | (4.7) | 34.3 | 65.3 | 75.9 | 86.4 |
| | | Coast | (0.9) | 29.8 | 60.8 | 75.0 | 93.7 |
| | | Ocean | (7.6) | 49.7 | 73.7 | 82.5 | 96.8 |
| | Snow-free (37.0) | Land | (5.4) | 8.0 | 25.6 | 68.5 | 97.7 |
| | | Desert | (2.6) | 8.2 | 23.5 | 55.8 | 95.4 |
| | | Coast | (0.9) | 10.9 | 23.0 | 60.9 | 96.4 |
| | | Ocean | (28.1) | 22.9 | 22.4 | 61.8 | 94.6 |

**Table 4**. Global cloud amount (%) calculated with different 'best guess' interpretations of the MODIS cloud mask product. Only MODIS IFOV collocated with CALIOP are considered.

| | CALIOP | MODIS cloud mask interpretation scenario | | |
| | | Only 'confident cloudy' is 'cloudy' | 'Confident clear' and 'probably clear' are clear, while the rest is cloudy | Only 'confident clear' is 'clear' |
|---|---|---|---|---|
| *Global* | | | | |
| Day+Night | 66.7 | 57.7 | 63.5 | 71.0 |
| Day | 64.3 | 59.3 | 63.4 | 70.6 |
| Night | 68.9 | 56.1 | 63.5 | 71.3 |
| *Polar regions (latitudes above 60ºN/S)* | | | | |
| Day+Night | 66.9 | 50.5 | 57.6 | 61.0 |
| Day | 64.8 | 59.0 | 62.6 | 66.4 |
| Night | 68.5 | 44.1 | 53.9 | 57.0 |
| *Equatorial region (latitudes between 30ºN and 30ºS)* | | | | |
| Day+Night | 59.8 | 52.8 | 58.0 | 67.4 |
| Day | 56.2 | 49.9 | 54.8 | 65.4 |
| Night | 63.5 | 55.7 | 61.2 | 69.4 |
| *Mid-latitudes (between polar and equatorial)* | | | | |
| Day+Night | 73.3 | 68.9 | 74.2 | 83.6 |
| Day | 72.0 | 69.1 | 72.7 | 79.1 |
| Night | 74.6 | 68.8 | 75.7 | 87.9 |

**Figures**


Figure 1. Observations declared 'confident clear' or 'probably clear' by the MODIS cloud masking algorithm, but identified as 'cloudy' by CALIOP.

Figure 2. Observations declared 'confident cloudy' or 'probably cloudy' by the MODIS cloud masking algorithm, but
identified as 'clear' by CALIOP.

Figure 3. CALIOP-based cloud fraction for MODIS cloud mask classes.

Figure 4. CALIOP-based cloud fraction for MODIS cloud mask classes for the 'nighttime, snow(ice)-covered ocean' algorithm
path, and corresponding histograms (red vertical line indicates the mean value).

Figure 5. CALIOP-based cloud fraction for MODIS cloud mask classes for the 'nighttime snow-free land' algorithm path, and corresponding histograms (red vertical line indicates the mean value).

Figure 6. CALIOP-based cloud fraction for MODIS cloud mask classes for the 'daytime, snow-free land' algorithm path, and corresponding histograms (red vertical line indicates the mean value).

Figure 7. CALIOP-based cloud fraction for MODIS cloud mask classes for the 'daytime, snow-free ocean' algorithm path, and corresponding histograms (red vertical line indicates the mean value).

Figure 8. Difference between the MODIS Science Team (MODIS ST) Level 3 cloud amount product, and cloud amount calculated with the cloud fractions found in this study. Positive values indicate that the MODIS operational product overestimates cloud amount (with respect to CALIOP), while negative values indicate a MODIS underestimate. All MODIS observations refer to the full swath, not only those collocated with CALIOP.

**Supplementary Online Material**

Figure S1. The average cloud-aerosol discrimination (CAD) score for CALIOP cloud data used in the study. Maps show spatial variation in the CAD score during the day (b), at night (c), and regardless of the time of the day (a). These plots demonstrate the high stability of CAD scores at various latitudes during the day (d) and at night (e).

    Figure S2. Overall accuracy of MODIS cloud detection as a function of the temporal (a, c) and spatial (b, d) separation of
MODIS and CALIOP IFOVs. Top plots show the frequency of observations for individual time (a) and distance (b) ranges, while bottom plots report accuracy for these ranges. MODIS detections are validated using CALIOP cloud profiles as a reference. Accuracy is defined as the ratio of MODIS true detections (true positive and true negative) to all MODIS observations (see Table 1 in the main text for details).

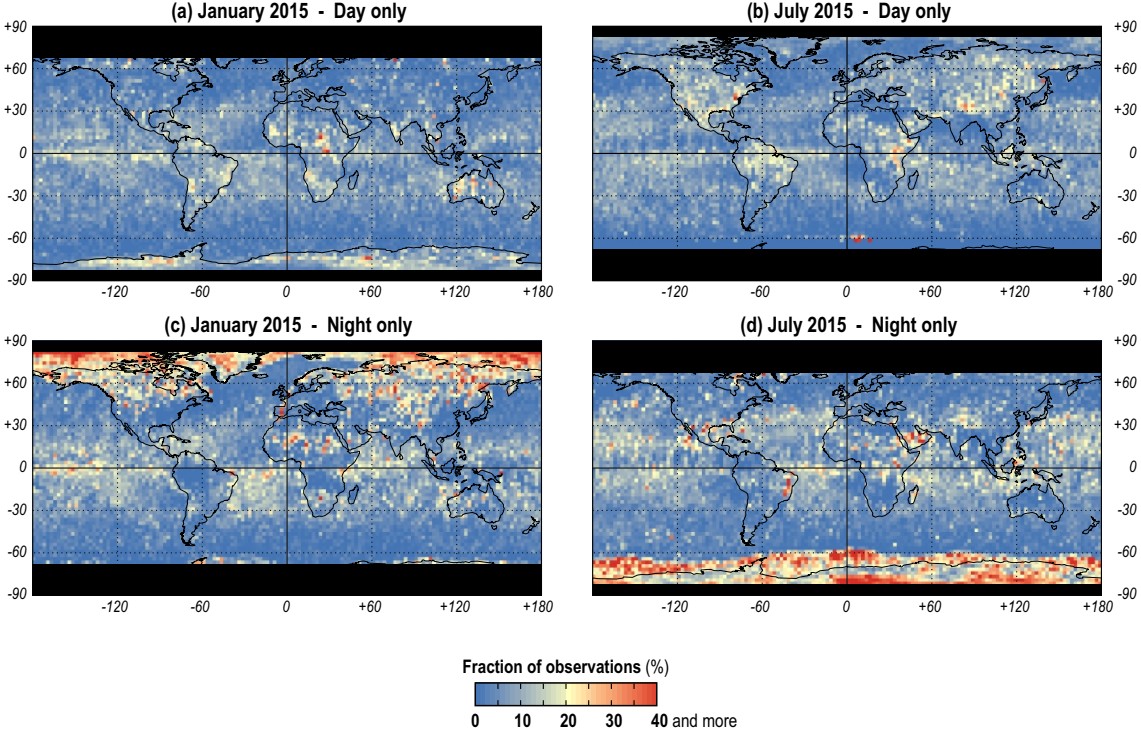

**Figure 1.** Observations declared 'confident clear' or 'probably clear' by the MODIS cloud masking algorithm, but identified as 'cloudy' by CALIOP.

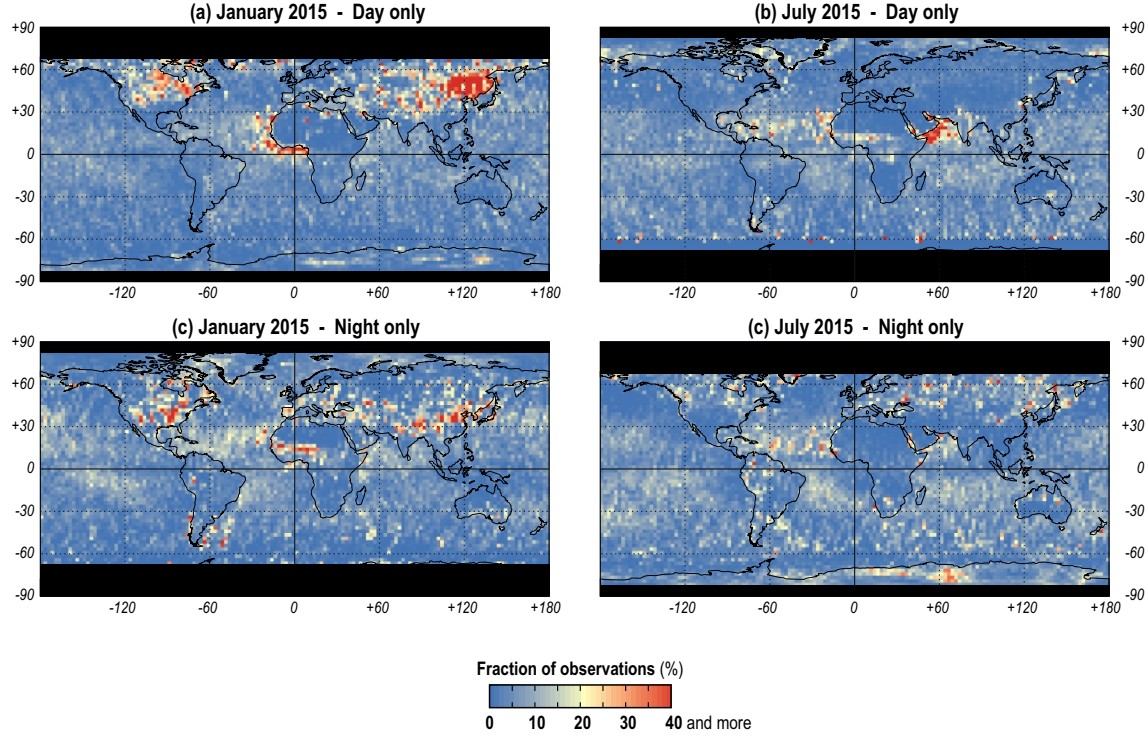

**Figure 2.** Observations declared 'confident cloudy' or 'probably cloudy' by the MODIS cloud masking algorithm, but identified as 'clear' by CALIOP.

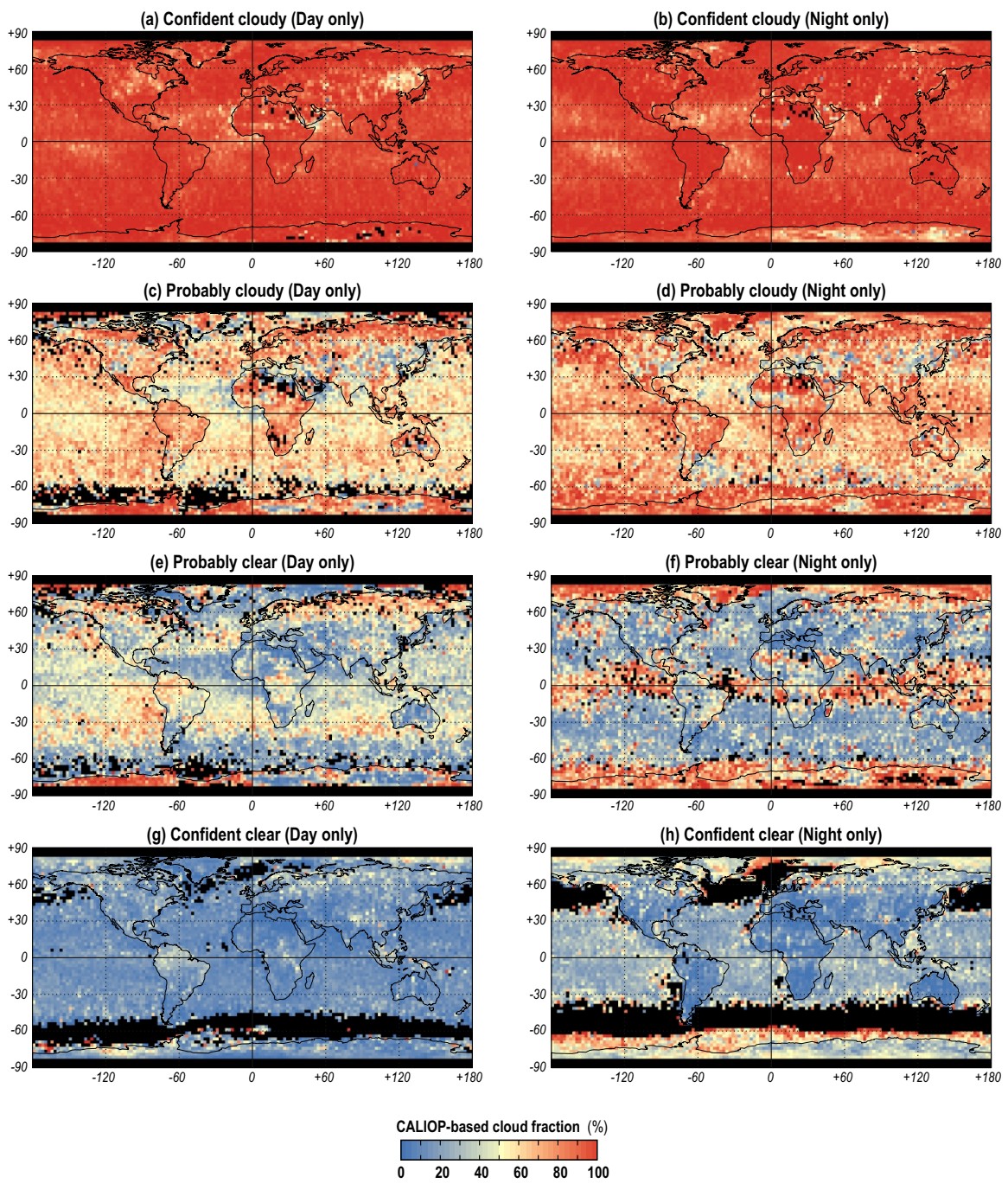

**Figure 3**.CALIOP-based cloud fraction for MODIS cloud mask classes.

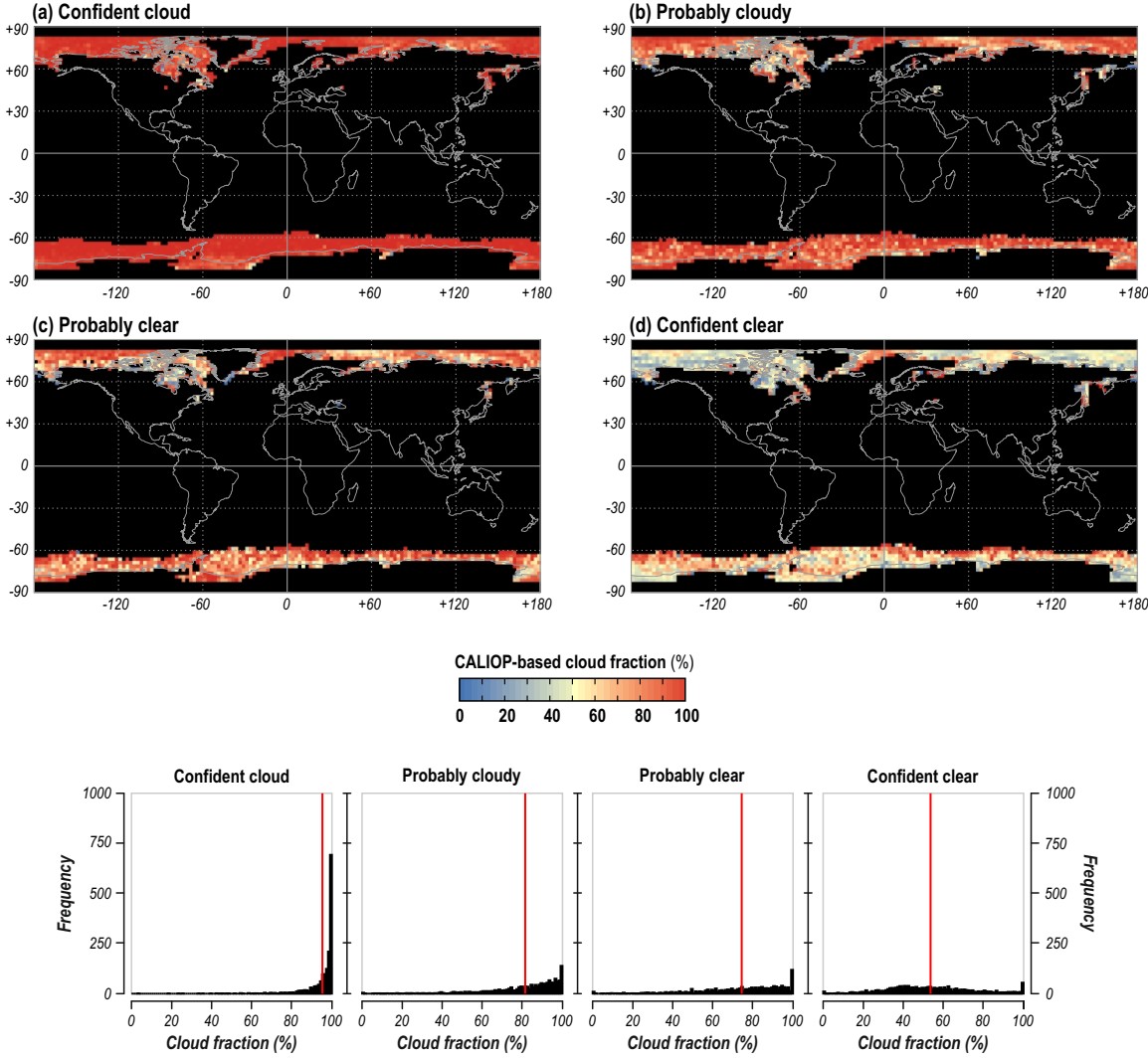

**Figure 4**. CALIOP-based cloud fraction for MODIS cloud mask classes for the 'nighttime, snow(ice)-covered ocean' algorithm path, and corresponding histograms (red vertical line indicates the mean value).

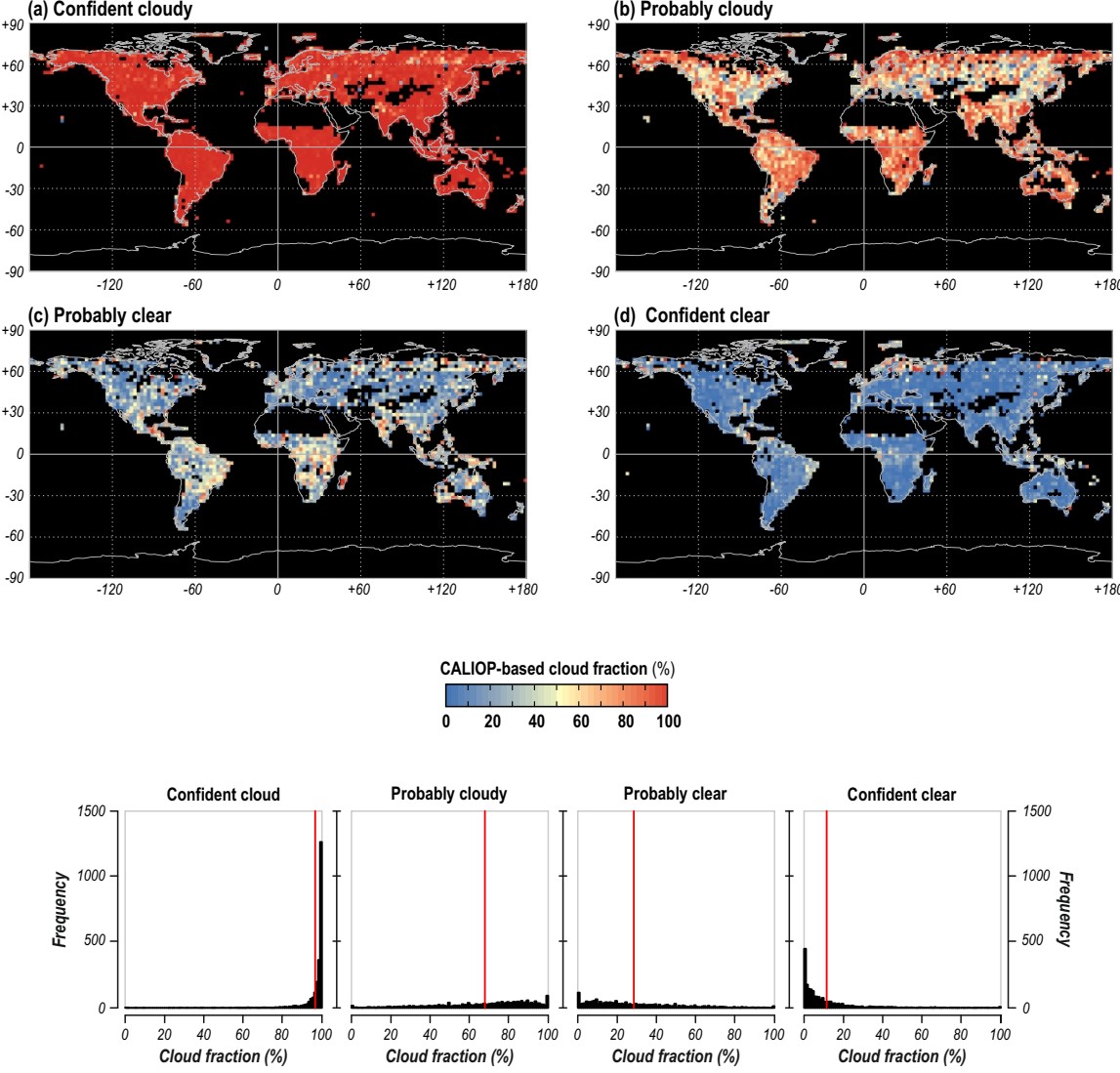

**Figure 5**. CALIOP-based cloud fraction for MODIS cloud mask classes for the 'nighttime snow-free land' algorithm path, and corresponding histograms (red vertical line indicates the mean value).

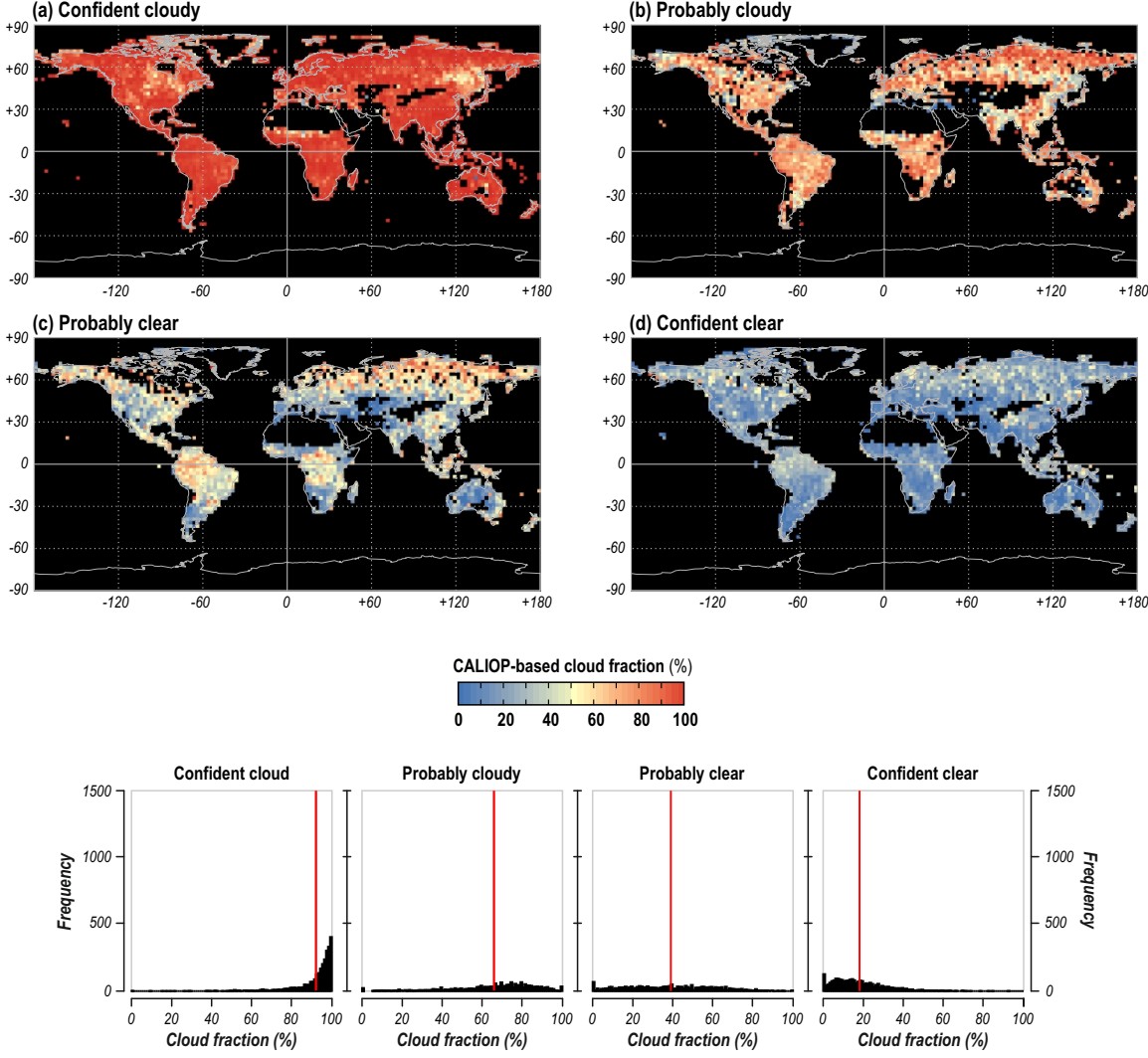

**Figure 6**. CALIOP-based cloud fraction for MODIS cloud mask classes for the 'daytime, snow-free land' algorithm path, and corresponding histograms (red vertical line indicates the mean value).

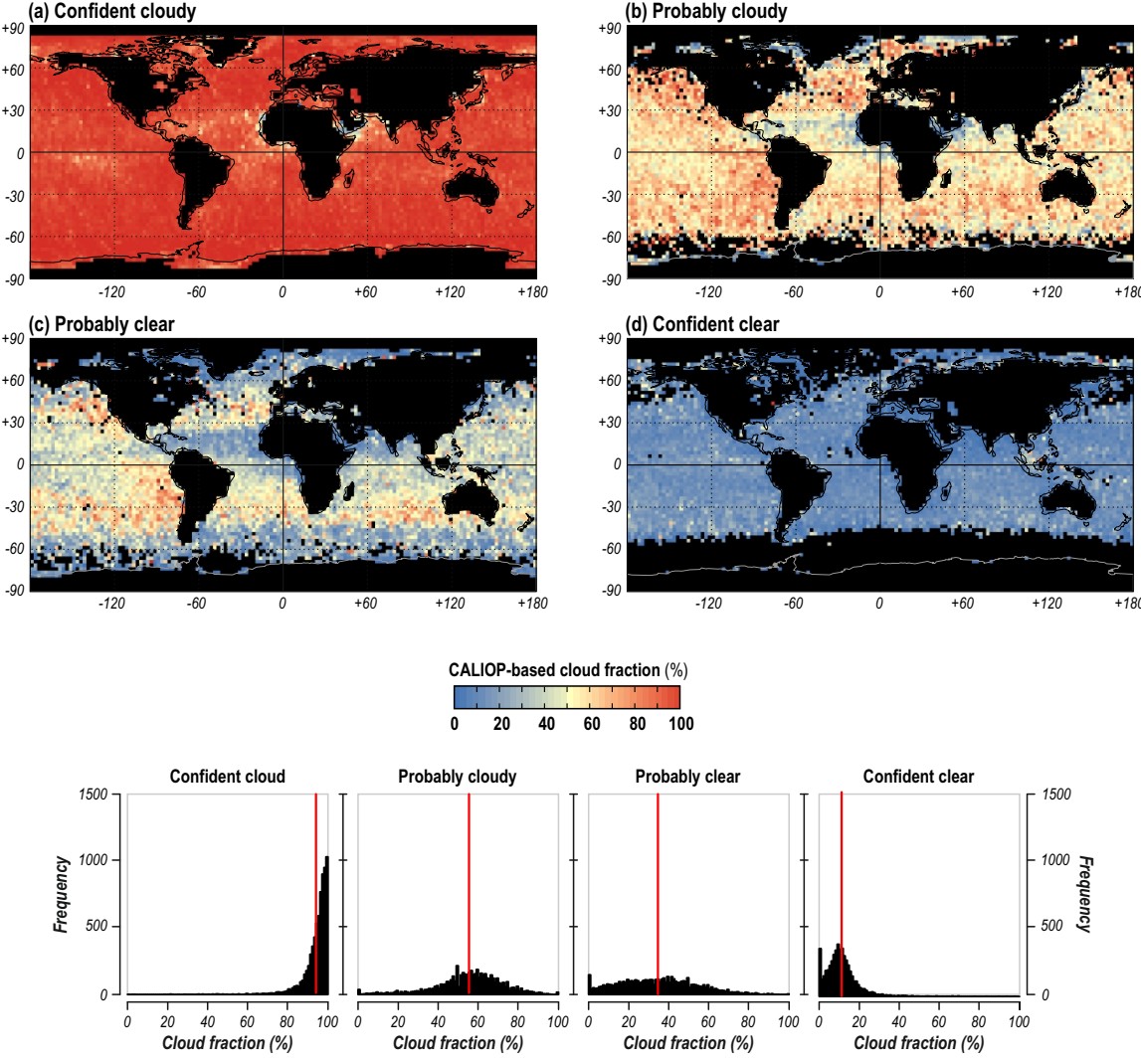

**Figure 7**. CALIOP-based cloud fraction for MODIS cloud mask classes for the 'daytime, snow-free ocean' algorithm path, and corresponding histograms (red vertical line indicates the mean value).

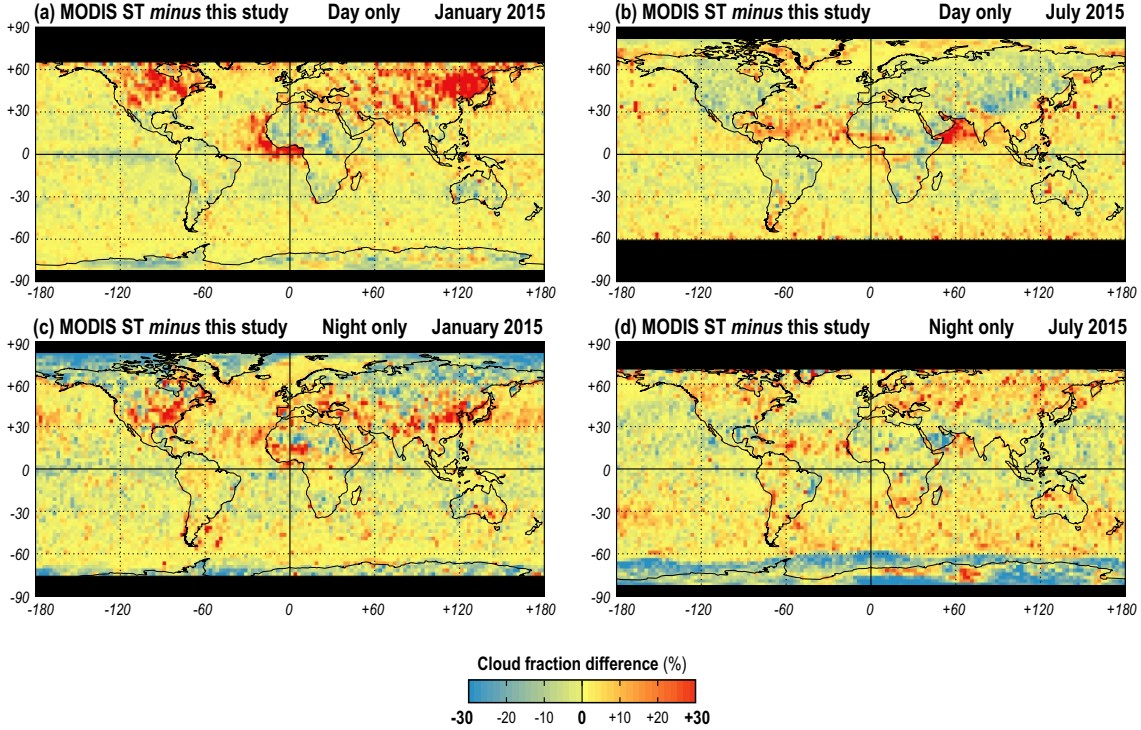

**Figure 8**. Difference between the MODIS Science Team (MODIS ST)  Level 3 cloud amount product, and cloud amount calculated with the cloud fractions found in this study. Positive values indicate that the MODIS operational product overestimates cloud amount (with respect to CALIOP), while negative values indicate a MODIS underestimate. All MODIS observations refer to the full swath, not only those collocated with CALIOP.