# Peer review of "Calibration of global MODIS cloud amount using CALIOP cloud profiles"

_Atmospheric Measurement Techniques, 2020_

## Short Comment (SC1) · 22 Apr 2020

I don't understand the NASA can assign 0% cloud cover to probably clear and 100% cloud cover to probably cloudy when the same values are assigned to the confidently cases. It seems more logical to assign 33% and 66% to these two categories respectively, since they did not pass all the tests required to be either confidently clear or confidently cloudy. It might be useful to know how such a change in their product would effect your results?

---

## Author Comment (AC1) · 27 Apr 2020

Thank you for the comment. Assigning 33% to 'probably clear' and 66% to 'probably cloud' would definitively change the final estimation of cloud amount. To get the exact figure all the data for this study would have to be reworked. However the existing results suggest that 66% is well justified (we found 66,6%), and 33% is only $\sim$5% more that 27.7% found by us. The difference remains for 'confident' classes, which are the most frequent and influence the final results significantly (on a global scale).

---

## Author Comment (AC2) · 27 Apr 2020

SC2 is a copy of SC1.

———————————————————

---

## Referee Comment (RC1) · Anonymous Referee #1 · 19 May 2020

Review of AMT-2020-111 Title: Calibration of global MODIS cloud amount using CALIOP cloud profiles Author: Andrzej Z. Kotarba

Comments to the Editor

The author gives a very detailed critique of the operational MODIS cloud mask (MOD35) aggregation strategy using the CALIOP lidar cloud detection product as "ground truth". Collocated 1-km CALIOP and Aqua MODIS data is used to assign mean cloud amounts to the four output cloud mask categories (confident clear, probably clear, probably cloudy, confident cloudy) for various illumination and surface types. The manuscript is very well written and organized with tables and figures that add detail and understanding to the text. The main object of the paper is to ascertain the suitability of the operational aggregation method from pixel level (Level 2) to tempo-

ral and spatial averages (Level 3). The current method simply compares numbers of cloudy pixels to the total number when calculating cloud amounts, where "cloudy" includes confident cloudy and probably cloudy categories and "clear" is confident clear and probably clear categories. This makes the implicit assumption that the two cloudy categories indicate 100% cloudiness while the two clear categories imply 0% cloudiness. The author concludes that this method leads to significant errors in regional level 3 cloud amounts and reassigns cloudiness values to each mask output category based on the collocated CALIOP cloud detection data.

My only objection is that the author makes the assumption that variability in the confidence of clear sky depends only on cloud fractions within 1-km pixels. However, there are other possibilities. Some of them are:

1) optically thin clouds that cover entire pixels (thin cirrus) 2) roughness of the ocean surface and wave orientation 3) orientation of clouds relative to the sun (scattering angle, 3-D effects) 4) variability in surface characteristics (brightness, topography, shadows, land/water boundaries)

The stated philosophy of the MODIS cloud mask is to be clear sky conservative (see cloud mask ATBD), i.e., if there is any hint of cloudiness in a given pixel, it should be considered "not clear". In this sense then, the practice of considering both confident and probably cloudy pixels to be "cloudy" during aggregation seems reasonable to me, given that no information about cloud morphology is available. However, this is not to say that the present study is not useful as an error analysis or otherwise beneficial to users.

I am recommending the manuscript be accepted for publication with minor revisions as outlined in the comments to the author.

I have one question for the editor. What is the convention for single-author papers - should the author use "we" or "I"? See line 13 of the abstract among other locations.

General Comments to the Author

The author gives a very detailed critique of the operational MODIS cloud mask (MOD35) aggregation strategy using the CALIOP lidar cloud detection product as "ground truth". Collocated 1-km CALIOP and Aqua MODIS data is used to assign mean cloud amounts to the four output cloud mask categories (confident clear, probably clear, probably cloudy, confident cloudy) for various illumination and surface types. The manuscript is very well written and organized with tables and figures that add detail and understanding to the text. The main object of the paper is to ascertain the suitability of the operational aggregation method from pixel level (Level 2) to temporal and spatial averages (Level 3). The current method simply compares numbers of cloudy pixels to the total number when calculating cloud amounts, where "cloudy" includes confident cloudy and probably cloudy categories and "clear" is confident clear and probably clear categories. This makes the implicit assumption that the two cloudy categories indicate 100% cloudiness while the two clear categories imply 0% cloudiness. The author concludes that this method leads to significant errors in regional level 3 cloud amounts and reassigns cloudiness values to each mask output category based on the collocated CALIOP cloud detection data.

My only objection is that the author makes the assumption that variability in the confidence of clear sky depends only on cloud fractions within 1-km pixels. However, there are other possibilities. Some of them are:

1) optically thin clouds that cover an entire pixel (thin cirrus) 2) surface brightness approaches that of clouds 3) orientation of clouds relative to the sun (scattering angle, 3-D effects) 4) variability in surface characteristics (brightness, topography, shadows, land/water boundaries)

The stated philosophy of the MODIS cloud mask is to be clear sky conservative (see cloud mask ATBD), i.e., if there is any hint of cloudiness in a given pixel, it should be considered "not clear". In this sense then, the practice of considering both confident

and probably cloudy pixels to be "cloudy" during aggregation seems reasonable to me, given that no information about cloud morphology is available. However, this is not to say that the present study is not useful as an error analysis or otherwise beneficial to users.

I am recommending the manuscript be accepted for publication with minor revisions as outlined in the specific comments below.

Specific Comments to the Author

Many high, very optically thin clouds detected by CALIOP have no chance to be categorized as cloudy by observations from a passive radiometer such as MODIS. Please mention this as a partial reason for the 21.5% cloud amount associated with confident clear.

Abstract: What is the meaning of "uncertainties were related to the efficiency of the cloud masking algorithm"? Please clarify or delete.

Please delete "Until the algorithm can be significantly modified". After 20 years, algorithm issues notwithstanding, large and potentially disruptive modifications to the cloud mask are unlikely and probably unwise.

Line 133: "IFOV" should be "scan lines". Line 183: "probably cloudy" should be "probably clear".

Line 140: The first two sentences beginning at line 140 are probably better placed at the beginning of Section 3.2. I would eliminate the last sentence of this section as it seems superfluous.

Lines 152-154: This would be a good place to insert a few words about the difficulty of cloud detection from passive instruments during polar night. Thermal contrast is almost nil in these situations and what does exist is often due to temperature inversions, many times multiple ones, that exist with or without clouds being present. Please explain that in polar night, CALIOP has an even bigger advantage in detecting clouds than in

warmer climes.

Line 258: What is "the most modest version" of the MODIS cloud mask? Please explain or eliminate the phrase.

Lines 265-268: Given the errors inherent in remote sensing of cloud properties in general, and in the difficulty of accurate cloud detection in particular, I am surprised that the author would ever expect 100% accuracy from any type of cloud amount calculation. All algorithms are inadequate in some way and to some extent. I strongly advise the author to eliminate the section beginning with "We found the approach to be inadequate" and ending with "environmental conditions". The statistics given here are just a restatement of previously reported results. Of course there are limitations to the cloud masking procedure and undoubtedly "certain cloud regimes and/or environmental conditions" are more difficult than others. On the other hand, it is quite fair to report the results of the 100% clear/cloud assumptions in the MODIS cloud amount calculations, as is done immediately following.

Line 275: I think you mean Table 3. Please add a sentence or two justifying the use of collocated near-nadir CALIOP data on entire swaths of MODIS data or a description of a corrective measure.

Line 310: The sentence beginning with "Therefore, the standard" is unnecessary.

Line 327: Variability is to be expected within algorithm paths as they are necessarily very general categories. The statement that the same thresholds are applied in widely varying locations is not completely true. The important 0.65 $\mu$m daytime land cloud test is a function of background NDVI and scattering angle. Please include this information.

Line 329: Again, the text "Until significant modifications are made to the MODIS cloud masking algorithm," is unnecessary and a bit high-handed. I would simply begin with "CALIOP-based . . .".

References Line 63: Fontana et al., 2013 is missing from the reference list.

Figures and Tables Table 3: Caption should indicate that the cloud fractions listed are CALIOP-based.

---

## Referee Comment (RC2) · Anonymous Referee #2 · 3 Jun 2020

**Calibration of global MODIS cloud amount using CALIOP cloud profiles**

Andrzej Z. Kotarba[1]

[1]Space Research Centre, Polish Academy of Sciences, 00-716 Warsaw, Poland

*Correspondence to*: Andrzej Z. Kotarba (akotarba@cbk.waw.pl)

**Reviewer comments:**

This paper identified a new set of cloud fractions corresponding to the four MODIS cloud masks using collocated CALIOP measurements. The author suggested using these new fractions to replace those currently used in the operational MODIS product to decrease uncertainties. Other than global assessments, the author further examined how those fractions changed for different MODIS cloud masking algorithm paths and at different latitude regions. The author recommended using local cloud fractions instead of global cloud fractions. As these cloud fractions are crucial to derive MODIS level 3 cloud products, the author compared the cloud amount differences in level 3 cloud products using new set and operational cloud fractions. It showed that using new set of cloud fractions successfully solved several issues of current L3 cloud product.

This work proposed a new set of cloud fractions to replace the current operational cloud fractions in MODIS cloud mask algorithm to improve current MODIS level 3 cloud product. The work is important for cloud climatology community. The reviewer recommended the paper for publication after some minor changes.

**Major comments:**

1. Line 71: the author mentioned that "no research-based, objective alternatives to the 0/0/100/100 interpretation currently in use have been put forward". However Line 250 reviewed the validation work conducted by Wang et al. (2016) which also estimated the cloud fraction for four MODIS cloud masks. The reviewer suggested reviewing Wang's work in the introduction and also emphasizing what's new in this work. For example, Wang's work focused on daytime only, this work included daytime, night time and both day and night time. Moreover, this work examined how those four cloud fractions changed for different MODIS cloud mask algorithm paths and latitude regions.

2. The cloud fractions were derived with two months data, i.e., January and July 2015. While the author demonstrated the fractions could have a large variability depending on environmental conditions. Could they also have a seasonal variation? How valid to apply the same numbers to different seasons for the whole MODIS mission?

3. The author considered CALIPSO data as "ground truth" by including all cloud layers detected by CALIOP. As CALIOP data reported quality flags, it is possible to choose confident clouds only. For example, including clouds with cloud-aerosol discrimination score between 20 and 100 (low, middle and high confidence) or 70 and 100 (high confidence only) by specifying the range of parameter CAD_Score. Not sure how this filter might change the current findings in the paper.

4. In the paper, the cloud fractions are further estimated for each cloud mask algorithm path and day/night conditions. It is noted that the CALIOP has different detection sensitivity during day and night, i.e., CALIOP is able to detect more thin cirrus clouds around the tropical region at night than during the day. This might help understand the day/night discrepancies in Figure1-3.

5. As briefly touched by the author in Line 238, the level 2 CALIOP cloud layer product reported detected cloud layers only. It is very possible there are aerosol layers detected and those aerosol layers would be reported in aerosol products but not in cloud products. In this scenario, the sky is not exactly "clear". To avoid confusions, some researchers use "cloud free" instead "clear".

**Minor comments:**

1. Abstract: keep consistency when describing four cloud fraction numbers and cloud mask categories. Line 7: "confident cloudy", "probably cloudy", "probably clear", "confident clear". Line 14: 21.5%, 27.7%, 66.6%, 94.7%.

2. Line 16: "selected locations"? Please give a few locations as examples.

3. Line 17: "error" → "uncertainty"?

4. Line 18: What is "our method"?

5. Line 19: "robust" is a strong word. Does the author would like to say something like "We recommend using the cloud fraction ratios found in this work to improve MODIS estimates."

6. Line 20: "other mission"? Other passive missions?

7. Line 24: "W m-2" should be "W m$^{-2}$".

8. Line 48: "The procedure implemented by NASA…" → The procedure implemented by MODIS science working group?

9. Line 51: "- see, for example, " → e.g. ?

10. Line 54: "NASA's approach" → standard procedure? It is not an approach from an agency. Instead, it is from MODIS science working group.

11. Line 54: "… are both allowed and in use." → "… are adopted by other groups." ?

12. Line 63: Moved "in Switzerland" after "observations". It would be nice to specify the number of ground-based observations, i.e., "… compared MODIS data with n ground ground-based observations…".

13. Line 70: "NASA standard approach" → standard procedure or standard approach?

14. Line 71: "… currently in use have been put forward" is confusing. Does the author mean "… currently widely used are still missing" or something like that?

15. Line 72: "… based on quantitative, empirical lidar observations" is confusing. Does the author mean "… based on a quantitative analysis with lidar observations"?

16. Line 75: The CALIPSO was launched in 2006 instead of 2016.

17. Line 77-78: Consider removing "This is because" and "which means that" to make a concise and formal statement.

18. Line 83: Add "with CALIOP observations" after "… correspond to".

19. Line 83: Again it is not an approach from an agency. The author probably meant "current standard approach" or "current standard procedure".

20. Line 84: Does the author mean "Finally, we evaluate whether the MODIS Level 3 standard approach is reliable"?

21. Line 101: Consider removing "This is made available".

22. Line 103: Consider replacing "product; this was used to assign" with "with".

23. Line 108: Below 8.2 km, CALIOP has a horizontal resolution 0.333 km not 0.33 km.

24. Line 109: Between 20.2 km and 30.1 km, CALIOP has a horizontal resolution 5/3 km and vertical resolution 180 m. From 30.1 km to 40 km, the horizontal resolution is 5 km and the vertical resolution is 300 m. Please refer to Table 2 in Winker et al. [2006].

25. Line 114: "CAL_LID_L2" → level 2 cloud layer products.

26. Line 115: (version 4.20) → (version 4.20, CAL_LID_L2_01kmCLay-Standard-V4-20)?

27. Line 119: "Number Layers Found" variable → "Number_Layers_Found" parameter

28. Line 130: "… January and July 2005 …" should be "… January and July 2015 …" Any special reasons to choose these two months?

29. Line 141: Add "MODIS" after "perfect" would help a reader understand.

30. Line 147: Based on Table 1, should the number "86.7%" be "64.2%" at night?

31. Line 151: Should the number "77.4%" be "73.3%"?

32. Line 157: Is this region "ITCZ"? Does this high frequency misdetections due to high sensitivity of CALIOP? In other words, CALIOP detected very thin cirrus clouds which are invisible to MODIS.

33. Line 159: "… MODIS tended to falsely detect cloud rather than fail to detect it". This sentence is confusing. Does this mean higher percentage occurrence or larger area spatial extent? Should "Only" be removed?

34. Line 166: It is not exactly "every fifth MODIS" even though the percentage is about 20%.

35. Line 172-173: "no significant day/night difference" even though it is 12.3% for 'probably cloud'?

36. Figure 3g and 3h: What does black color over Southern Ocean mean?

37. Line 183: Should 'probably cloudy' be 'probably clear'?

38. Line 186: What does "this" in '…, but this was …" mean?

39. Table 3: Use same terms to describe snow-covered conditions in the context and table caption. For example, use "Snow-free" and "Snow-covered" or "No snow" and "Snow".

40. Line 205- 215: The author chose three cloud masking algorithm paths for detailed discussion. It would help a reader understand why those three if providing some explanations. Explain "Results" in Line 205 and "A similar pattern" in Line 211. Which results? Which pattern?

41. Line 223: Add a dot between MODIS collection "6" and "1"?

42. Line 225: It is confusing to discuss level 3 product here since no plots or work on level 3 clouds presented so far.

43. Line 235 and Line 240: The author claimed that temporal and spatial separations between Aqua and CALIPSO do not impact the results significantly. If not complicated, it is a good idea to show the plots when using different time and range shifts.

44. Line 246: Explain acronym "AVHRR".

45. Line 316: What is the spatial grid used to plot Figure 8?

46. Line 321: The author drew a conclusion "Whenever MODIS cloud amount is estimated at a spatial resolution of ~10 degrees of finer, …". There seems no evidence in the paper to support this conclusion. Something missing?

47. Line 324: Discussions on MODIS level 3 cloud product could be moved from "Summary and Conclusions" section to previous "Discussion" section.

---

## Author Comment (AC3) · 10 Jul 2020

**Reply to: Anonymous Referee #1**

**General Comments to the Author**

The author gives a very detailed critique of the operational MODIS cloud mask (MOD35) aggregation strategy using the CALIOP lidar cloud detection product as "ground truth". Collocated 1-km CALIOP and Aqua MODIS data is used to assign mean cloud amounts to the four output cloud mask categories (confident clear, probably clear, probably cloudy, confident cloudy) for various illumination and surface types.

The manuscript is very well written and organized with tables and figures that add detail and understanding to the text. The main object of the paper is to ascertain the suitability of the operational aggregation method from pixel level (Level 2) to temporal and spatial averages (Level 3). The current method simply compares numbers of cloudy pixels to the total number when calculating cloud amounts, where "cloudy" includes confident cloudy and probably cloudy categories and "clear" is confident clear and probably clear categories. This makes the implicit assumption that the two cloudy categories indicate 100% cloudiness while the two clear categories imply 0% cloudiness. The author concludes that this method leads to significant errors in regional level 3 cloud amounts and reassigns cloudiness values to each mask output category based on the collocated CALIOP cloud detection data.

My only objection is that the author makes the assumption that variability in the confidence of clear sky depends only on cloud fractions within 1-km pixels. However, there are other possibilities. Some of them are: 1) optically thin clouds that cover an entire pixel (thin cirrus) 2) surface brightness approaches that of clouds 3) orientation of clouds relative to the sun (scattering angle, 3-D effects) 4) variability in surface characteristics (brightness, topography, shadows, land/water boundaries).

The stated philosophy of the MODIS cloud mask is to be clear sky conservative (see cloud mask ATBD), i.e., if there is any hint of cloudiness in a given pixel, it should be considered "not clear". In this sense then, the practice of considering both confident and probably cloudy pixels to be "cloudy" during aggregation seems reasonable to me, given that no information about cloud morphology is available. However, this is not to say that the present study is not useful as an error analysis or otherwise beneficial to users.

I am recommending the manuscript be accepted for publication with minor revisions as outlined in the specific comments below.

*I am aware that the confidence in detecting clear sky is not only a function of cloud fraction within the IFOV. Thermal and reflectance contrast between a cloud and a background is controlled by many factors, including those listed by the Reviewer. The role of the cloud masking algorithm is to account for all of these factors as closely as possible, and maximize cloud detection success. In this study, only the resulting product of cloud detection (the Level 2 cloud mask) is evaluated. There is no attempt to investigate which factor – and to what degree – impacts the performance of the L2 algorithm. It is assumed that the algorithm, and the resulting L2 product, come "as they are", and are not 100% perfect.*

*The goal is to use the MODIS L2 product with independent data (namely CALIOP) to evaluate how the further processing of Level 2 observations impacts the uncertainty level of the L3 product (gridded monthly cloud amount). The study does not focus on generating the Level 2 product itself, but on calculating the Level 3 data, which is why cloud detection conditions are not explored in detail. We only consider day-time and night-time differences, latitudinal variability, and individual algorithm paths – assuming the latter reflect variation in the background's brightness temperature, reflectance, topography, etc. Differences in MODIS and CALIOP sensitivity to cirrus are discussed.*

**Specific Comments to the Author**

Many high, very optically thin clouds detected by CALIOP have no chance to be categorized as cloudy by observations from a passive radiometer such as MODIS. Please mention this as a partial reason for the 21.5% cloud amount associated with confident clear.

*Explanation added (in the Discussion section), as suggested.*

Abstract: What is the meaning of "uncertainties were related to the efficiency of the cloud masking algorithm"? Please clarify or delete.

*This has been deleted in order to keep the abstract concise (clarification at this point would make the Abstract too discursive).*

Please delete "Until the algorithm can be significantly modified". After 20 years, algorithm issues notwithstanding, large and potentially disruptive modifications to the cloud mask are unlikely and probably unwise.

*Deleted, as suggested.*

Line 133: "IFOV" should be "scan lines".

*In my opinion, 'IFOV' is the correct term. The CALIOP-MODIS matching procedure is IFOV-based, not scan line-based. MODIS is a whiskbroom-type scanner, meaning one rotation of the instrument's mirror results in a scan of 10 lines (considering 1 km detections only). Each line is then divided into 1354 instantaneous fields of view. Only a few, those located close to the MODIS ground track (~10 IFOVs per scan event, ~2030 per data granule) can be matched with CALIOP detections. The use of 'scan lines' would be misleading in this context.*

Line 183: "probably cloudy" should be "probably clear".

*Corrected.*

Line 140: The first two sentences beginning at line 140 are probably better placed at the beginning of Section 3.2. I would eliminate the last sentence of this section as it seems superfluous.

*Changed, as suggested.*

Lines 152-154: This would be a good place to insert a few words about the difficulty of cloud detection from passive instruments during polar night. Thermal contrast is almost nil in these situations and what does exist is often due to temperature inversions, many times multiple ones, that exist with or without clouds being present. Please explain that in polar night, CALIOP has an even bigger advantage in detecting clouds than in warmer climes.

*The discussion has been added – as suggested – however, not in the Results, but in the Discussion section.*

Line 258: What is "the most modest version" of the MODIS cloud mask? Please explain or eliminate the phrase.

*Clarified, as suggested (changed to Collection 061).*

Lines 265-268: Given the errors inherent in remote sensing of cloud properties in general, and in the difficulty of accurate cloud detection in particular, I am surprised that the author would ever expect 100% accuracy from any type of cloud amount calculation. All algorithms are inadequate in some way

and to some extent. I strongly advise the author to eliminate the section beginning with "We found the approach to be inadequate" and ending with "environmental conditions". The statistics given here are just a restatement of previously reported results. Of course there are limitations to the cloud masking procedure and undoubtedly "certain cloud regimes and/or environmental conditions" are more difficult than others. On the other hand, it is quite fair to report the results of the 100% clear/cloud assumptions in the MODIS cloud amount calculations, as is done immediately following.

*The MODIS cloud detection algorithm is only one of many methods, and no method is completely free of limitations. 100% and 0% are only points of reference – the theoretical cloud fraction that an ideal, perfect algorithm would give (if it existed). However, because the Reviewer found this paragraph to be a restatement of previously-reported results, I have followed the recommendation and deleted the suggested part.*

Line 275: I think you mean Table 3. Please add a sentence or two justifying the use of collocated near-nadir CALIOP data on entire swaths of MODIS data or a description of a corrective measure.

*Corrected and additional information added.*

Line 310: The sentence beginning with "Therefore, the standard" is unnecessary.

*Deleted, as suggested.*

Line 327: Variability is to be expected within algorithm paths as they are necessarily very general categories. The statement that the same thresholds are applied in widely varying locations is not completely true. The important 0.65 µm daytime land cloud test is a function of background NDVI and scattering angle. Please include this information.

*Information included, as suggested.*

Line 329: Again, the text "Until significant modifications are made to the MODIS cloud masking algorithm," is unnecessary and a bit high-handed. I would simply begin with "CALIOP-based : : :".

*Changed, as suggested.*

References Line 63: Fontana et al., 2013 is missing from the reference list.

*Reference added.*

Figures and Tables Table 3: Caption should indicate that the cloud fractions listed are CALIOP-based.

*Information added, as suggested.*

---

## Author Comment (AC4) · 10 Jul 2020

**Reply to: Anonymous Referee #2**

**Major comments**:

1. Line 71: the author mentioned that "no research-based, objective alternatives to the 0/0/100/100 interpretation currently in use have been put forward". However Line 250 reviewed the validation work conducted by Wang et al. (2016) which also estimated the cloud fraction for four MODIS cloud masks. The reviewer suggested reviewing Wang's work in the introduction and also emphasizing what's new in this work. For example, Wang's work focused on daytime only, this work included daytime, night time and both day and night time. Moreover, this work examined how those four cloud fractions changed for different MODIS cloud mask algorithm paths and latitude regions.

*Wang et al. (2016) is an excellent study. It validates the MODIS cloud mask (daytime only), with a focus on multilayer clouds, and considering different cloud regimes (with 2D histograms). However, the latter study does not provide a CALIOP-based cloud fraction for each of the four MODIS cloud mask classes. It may be possible to obtain these fractions (global values only) by an analysis of the confusion matrix (Table 2) presented by the authors. However, no direct information is provided about these values, how such statistics could be derived, or why (while this is the main objective of our study). For this reason, I prefer not to change the Introduction. I do, however, fully acknowledge the work of Wang et al. (2016), and refer to their results in the Discussion.*

2. The cloud fractions were derived with two months data, i.e., January and July 2015. While the author demonstrated the fractions could have a large variability depending on environmental conditions. Could they also have a seasonal variation? How valid to apply the same numbers to different seasons for the whole MODIS mission?

*The seasonality of the CALIPSO-based cloud fraction for MODIS cloud mask classes can be expected wherever environmental conditions are dominated by a strong seasonal cycle, in particular regions where the cloud regime changes noticeably. On the other hand, seasonal environmental change is consistent with changes in the frequency of per-location MODIS algorithm paths. Therefore, when regional CALIOP-based fractions per algorithm path are used (instead of fixed global fractions) the seasonality effect is balanced (at least partially). An operational use of CALIPSO-based factions would require the development of a relevant 'climatology'. An investigation of such a climatology would be an interesting extension of this study.*

3. The author considered CALIPSO data as "ground truth" by including all cloud layers detected by CALIOP. As CALIOP data reported quality flags, it is possible to choose confident clouds only. For example, including clouds with cloud-aerosol discrimination score between 20 and 100 (low, middle and high confidence) or 70 and 100 (high confidence only) by specifying the range of parameter CAD_Score. Not sure how this filter might change the current findings in the paper.

*Our study found that 95.6% of analysed CALIOP observations had CAD confidence of at least 70%, and confidence was below 20% for only 1.5% of data. These statistics did not differ between day and night, or January and July. High, stable CAD values makes it possible to conclude that filtering for data with CAD >70% or >80% would have no impact on the results. On the other hand, CAD results varied slightly more in the tropics, and this issue is discussed in the paper.*

4. In the paper, the cloud fractions are further estimated for each cloud mask algorithm path and day/night conditions. It is noted that the CALIOP has different detection sensitivity during day and night, i.e., CALIOP is able to detect more thin cirrus clouds around the tropical region at night than during the day. This might help understand the day/night discrepancies in Figure1-3.

*Additional information about CALIOP daytime/ night-time sensitivity has been added to the Discussion.*

5. As briefly touched by the author in Line 238, the level 2 CALIOP cloud layer product reported detected cloud layers only. It is very possible there are aerosol layers detected and those aerosol layers would be reported in aerosol products but not in cloud products. In this scenario, the sky is not exactly "clear". To avoid confusions, some researchers use "cloud free" instead "clear".

*I agree that 'cloud-free' is much more accurate in the context of this research, and I have changed 'clear' to 'cloud-free' whenever possible. Nonetheless, I have retained 'clear' in the name of the MODIS cloud mask, since these names are widely (and officially) used in MODIS product documentation.*

**Minor comments:**

1. Abstract: keep consistency when describing four cloud fraction numbers and cloud mask categories. Line 7: "confident cloudy", "probably cloudy", "probably clear", "confident clear". Line 14: 21.5%, 27.7%, 66.6%, 94.7%.

*Corrected, as suggested.*

2. Line 16: "selected locations"? Please give a few locations as examples.

*Examples added, as suggested.*

3. Line 17: "error" → "uncertainty"?

*Changed – 'uncertainty' is the more relevant term.*

4. Line 18: What is "our method"?

*The method used in the study to calibrate MODIS cloud amount. This sentence has been rephrased.*

5. Line 19: "robust" is a strong word. Does the author would like to say something like "We recommend using the cloud fraction ratios found in this work to improve MODIS estimates."

*The sentence has been rephrased.*

6. Line 20: "other mission"? Other passive missions?

*Passive cloud imagers – the sentence has been rephrased.*

7. Line 24: "W m-2" should be "W m$^{-2}$".

*Corrected.*

8. Line 48: "The procedure implemented by NASA…" → The procedure implemented by MODIS science working group?

*I agree. The procedure was developed by the MODIS Science Team or – more precisely – the Atmosphere Discipline Group within the MODIS Science Team. As the MODIS Science Team is a collaboration coordinated by NASA, I used NASA, but I agree that MODIS Science Team is more accurate. NASA has been changed to MODIS Science Team throughout the manuscript.*

9. Line 51: "- see, for example, " → e.g. ?

*Changed.*

10. Line 54: "NASA's approach" → standard procedure? It is not an approach from an agency. Instead, it is from MODIS science working group.

*Changed. See reply to comment 8.*

11. Line 54: "… are both allowed and in use." → "… are adopted by other groups." ?

*Changed (shortened) to: "… are in use."*

12. Line 63: Moved "in Switzerland" after "observations". It would be nice to specify the number of ground-based observations, i.e., "… compared MODIS data with n ground ground-based observations…".

*Changed, as suggested.*

13. Line 70: "NASA standard approach" → standard procedure or standard approach?

*Changed to 'procedure'.*

14. Line 71: "… currently in use have been put forward" is confusing. Does the author mean "… currently widely used are still missing" or something like that?

*Rephrased.*

15. Line 72: "… based on quantitative, empirical lidar observations" is confusing. Does the author mean "… based on a quantitative analysis with lidar observations"?

*Rephrased.*

16. Line 75: The CALIPSO was launched in 2006 instead of 2016.

*Corrected.*

17. Line 77-78: Consider removing "This is because" and "which means that" to make a concise and formal statement.

*The sentence justifies why the study uses CALIOP as a reference. The phrase, "Furthermore, the use of short..." at the beginning of the following sentence is a logical continuation. Therefore, I prefer to leave the paragraph as it is.*

18. Line 83: Add "with CALIOP observations" after "… correspond to".

*Added, as suggested.*

19. Line 83: Again it is not an approach from an agency. The author probably meant "current standard approach" or "current standard procedure".

*Changed. See also reply to comment 8.*

20. Line 84: Does the author mean "Finally, we evaluate whether the MODIS Level 3 standard approach is reliable"?

*Clarified, as suggested.*

21. Line 101: Consider removing "This is made available".

*Rephrased, as suggested.*

22. Line 103: Consider replacing "product; this was used to assign" with "with".

*Rephrased, as suggested.*

23. Line 108: Below 8.2 km, CALIOP has a horizontal resolution 0.333 km not 0.33 km.

*'0.33 km' corrected to '0.333 km'*

24. Line 109: Between 20.2 km and 30.1 km, CALIOP has a horizontal resolution 5/3 km and vertical resolution 180 m. From 30.1 km to 40 km, the horizontal resolution is 5 km and the vertical resolution is 300 m. Please refer to Table 2 in Winker et al. [2006].

*Corrected and clarified, as suggested.*

25. Line 114: "CAL_LID_L2" → level 2 cloud layer products.

*Changed, as suggested.*

26. Line 115: (version 4.20) → (version 4.20, CAL_LID_L2_01kmCLay-Standard-V4-20)?

*Product codename added, as suggested.*

27. Line 119: "Number Layers Found" variable → "Number_Layers_Found" parameter

*Changed, as suggested.*

28. Line 130: "… January and July 2005 …" should be "… January and July 2015 …" Any special reasons to choose these two months?

*Yes, these two months represent atmospheric conditions for summer (July) and winter (January) in the northern hemisphere. The selection of these months makes it possible to investigate contrasting cloud regimes in mid-latitudes (more cumuliform in summer, more stratiform in winter) and season-dependent conditions for cloud detection (e.g. snow cover).*

29. Line 141: Add "MODIS" after "perfect" would help a reader understand.

*Added, as suggested.*

30. Line 147: Based on Table 1, should the number "86.7%" be "64.2%" at night?

*In fact, it should be '84.2% at night' (as in Table 1) – corrected.*

31. Line 151: Should the number "77.4%" be "73.3%"?

*Corrected.*

32. Line 157: Is this region "ITCZ"? Does this high frequency misdetections due to high sensitivity of CALIOP? In other words, CALIOP detected very thin cirrus clouds which are invisible to MODIS.

*Yes, it is the intertropical convergence zone. I have expanded on cloud detection by MODIS and CALIOP at low latitudes in the Discussion.*

33. Line 159: "… MODIS tended to falsely detect cloud rather than fail to detect it". This sentence is confusing. Does this mean higher percentage occurrence or larger area spatial extent? Should "Only" be removed?

*The statement was deleted.*

34. Line 166: It is not exactly "every fifth MODIS" even though the percentage is about 20%.

*Changed to "one fifth of MODIS".*

35. Line 172-173: "no significant day/night difference" even though it is 12.3% for 'probably cloud'?

*Clarified.*

36. Figure 3g and 3h: What does black color over Southern Ocean mean?

*It means there were no confident clear detections by MODIS in these regions at that time.*

37. Line 183: Should  probably cloudy' be 'probably clear'?

*Corrected – 'probably clear' is the correct term.*

38. Line 186: What does "this" in '…, but this was …" mean?

*Rephrased and clarified.*

39. Table 3: Use same terms to describe snow-covered conditions in the context and table caption. For example, use "Snow-free" and "Snow-covered" or "No snow" and "Snow".

*Corrected, as suggested ('snow-covered' and 'snow-free' are now used consistently).*

40. Line 205- 215: The author chose three cloud masking algorithm paths for detailed discussion. It would help a reader understand why those three if providing some explanations. Explain "Results" in Line 205 and "A similar pattern" in Line 211. Which results? Which pattern?

*Four algorithm paths are described in the text. The first is "the combination of night, an oceanic background and snow-cover (or sea ice)". This scenario is notable because it "constituted the 'most cloudy' scenario". The second is "snow-free land at night", this was chosen because: "Results [for it] were most consistent with the standard Level 3" (already mentioned in the manuscript). The two other scenarios are "snow-free land during the day", and "ice-free oceans". The choice of the latter is justified in the paper: it is "the most frequent algorithm path". I agree that the justification of the choice of "snow-free land during the day" was missing. Therefore, following the Reviewer's suggestion, I have added an explanation (it is of particular interest for land/ vegetation MODIS remote sensing).*

*Lines 205 and 211 have been clarified, as suggested.*

41. Line 223: Add a dot between MODIS collection "6" and "1"?

*There are two conventions in use: a three-digit name with leading zero (005, 055, 006, 061, etc.), or to divide a collection number by 100 and use a coma (5.0, 5.5, 6.0, 6.1, etc.). I prefer to use the first, hence '61' has been changed to '061'.*

42. Line 225: It is confusing to discuss level 3 product here since no plots or work on level 3 clouds presented so far.

*Clarified. The implications for Level-3 data are presented in the Discussion, but not before. The first paragraph of the section only introduces issues that are discussed in the following paragraphs. I have made this point clearer in the new version of the manuscript.*

43. Line 235 and Line 240: The author claimed that temporal and spatial separations between Aqua and CALIPSO do not impact the results significantly. If not complicated, it is a good idea to show the plots when using different time and range shifts.

*I have prepared the plots, as suggested. I also agree that they might be interesting for some readers. However, I leave it to the Editor to decide whether they should be included in the main text, or as additional/ supporting online material (the latter would be my choice).*

44. Line 246: Explain acronym "AVHRR".

*Explained, as suggested.*

45. Line 316: What is the spatial grid used to plot Figure 8?

*All figures use the equirectangular projection with 2.5°×2.5° spatial resolution.*

46. Line 321: The author drew a conclusion "Whenever MODIS cloud amount is estimated at a spatial resolution of ~10 degrees of finer, …". There seems no evidence in the paper to support this conclusion. Something missing?

*Ten degrees longitude/ latitude was the approximate area of cloud amount uncertainties in China, along the coast of the Arabian Peninsula, north-west Africa, and some locations in North America. However, I agree that the figure could be misleading when considering, for example, polar regions where the area is much larger. Consequently, the reference to "10 degrees" has been deleted, and replaced by "regional/local".*

47. Line 324: Discussions on MODIS level 3 cloud product could be moved from "Summary and Conclusions" section to previous "Discussion" section

*I prefer not to move the discussion about Level 3 data from the Discussion to the Results. The key 'technical' objective of the study was to derive CALIOP-based cloud fraction from MODIS. The outcome of this work is reported in the Results section. A discussion of the implications of these results for calculating global cloud amounts is a different matter. In my opinion, the present structure of the manuscript clearly separates the results of the study's calculations from a discussion of their impact.*